# Impact of DNA methylation on 3D genome structure

Diana Buitrago [1,2,7,9], Mireia Labrador [1,2,9], Juan Pablo Arcon [1,2], Rafael Lema[1,2], Oscar Flores[1,2], Anna Esteve-Codina [3,4], Julie Blanc [3,4], Nuria Villegas [1,2], David Bellido[5], Marta Gut [3,4], Pablo D. Dans[1,2,8], Simon C. Heath [3,4], Ivo G. Gut[3,4], Isabelle Brun Heath [1,2✉] & Modesto Orozco [1,2,6✉]

Determining the effect of DNA methylation on chromatin structure and function in higher organisms is challenging due to the extreme complexity of epigenetic regulation. We studied a simpler model system, budding yeast, that lacks DNA methylation machinery making it a perfect model system to study the intrinsic role of DNA methylation in chromatin structure and function. We expressed the murine DNA methyltransferases in *Saccharomyces cerevisiae* and analyzed the correlation between DNA methylation, nucleosome positioning, gene expression and 3D genome organization. Despite lacking the machinery for positioning and reading methylation marks, induced DNA methylation follows a conserved pattern with low methylation levels at the 5' end of the gene increasing gradually toward the 3' end, with concentration of methylated DNA in linkers and nucleosome free regions, and with actively expressed genes showing low and high levels of methylation at transcription start and terminating sites respectively, mimicking the patterns seen in mammals. We also see that DNA methylation increases chromatin condensation in peri-centromeric regions, decreases overall DNA flexibility, and favors the heterochromatin state. Taken together, these results demonstrate that methylation intrinsically modulates chromatin structure and function even in the absence of cellular machinery evolved to recognize and process the methylation signal.

[1] Institute for Research in Biomedicine (IRB Barcelona) - The Barcelona Institute of Science and Technology (BIST), Barcelona, Spain. [2] Joint IRB-BSC Program in Computational Biology, Barcelona, Spain. [3] CNAG-CRG, Centre for Genomic Regulation (CRG), Barcelona Institute of Science and Technology (BIST), Barcelona, Spain. [4] Universitat Pompeu Fabra (UPF), Barcelona, Spain. [5] Centres Cientifics i Tecnologics, Universitat de Barcelona, Barcelona, Spain. [6] Departament de Bioquímica i Biomedicina, Universitat de Barcelona, Barcelona, Spain. [7] Present address: Departamento de Física y Matemáticas, Universidad Autónoma de Manizales, Manizales, Colombia. [8] Present address: Department of Biological Sciences, CENUR North Coast. University of the Republic, Salto, Uruguay. [9] These authors contributed equally: Diana Buitrago, Mireia Labrador. ✉email: isabelle.heath@irbbarcelona.org; modesto.orozco@irbbarcelona.org

DNA methylation is one of the most important epigenetic marks which introduce major changes in cellular functionality, some of them with systemic impact and coupled to important pathologies in mammals (for a review see[1]). For example, mutations in DNA Methyltransferase 3b (*DNMT3b*) are implicated in Immunodeficiency, Centromere instability and Facial anomalies (ICF) syndrome[2], mutations in DNA Methyltransferase 3a (*DNMT3a*) are found in acute myeloid leukemia (AML) patients[3] while those in DNA Methyltransferase 1 (*DNMT1*) cause autosomal dominant cerebellar ataxia, deafness and narcolepsy[4]. DNA methylation plays a key role in development (reviewed in[5]) and cell differentiation[6], and a correct methylation level is crucial for the regulation of parental imprinting and X chromosome inactivation mechanisms[7]. Not surprisingly, changes in DNA methylation patterns have been associated with many different types of cancer in humans[8–11] and reviewed in[12,13].

It is commonly stated that DNA methylation in the promoter region of a gene is a hallmark of repression[1,14]. However, several studies have shown that DNA methylation in the gene body could also affect gene expression, and that the increase in methylation in promoter regions is not always correlated with gene repression, making the effect of DNA methylation on gene expression far more complicated than a simple on/off signal[15,16]. Two possible mechanisms might link DNA methylation with gene regulation: i) a direct effect involving proteins with methylated DNA binding domains, which act as anchoring points for other effector proteins regulating gene activity and ii) an indirect effect related to changes in DNA properties. In the second case, at least two possibilities emerge: i) DNA methylation changing the binding affinity of transcription factors to DNA and ii) DNA methylation affecting chromatin structure, which in turn modulates gene expression[17]. Accurate in silico and in vitro studies have demonstrated that DNA methylation makes the DNA less flexible and less likely to form nucleosomes[18,19], but other studies relying on in vitro nucleosome reconstitution claim the opposite: i.e., that DNA methylation increases the affinity of histones for DNA[20] and DNA methylation promotes compaction on pre-assembled nucleosomes[21]. Experiments using in vitro reconstituted nucleosomes and in vivo studies on mammals and plants are also confusing, with some suggesting that methylation occurs preferentially on nucleosomal DNA[22–24] and others concluding the opposite[25–28]. This controversy can probably be explained by the high complexity of mammalian genomes, with a myriad of factors controlling directly or indirectly nucleosome positioning[29], and by the existence of higher orders of chromatin structure beyond the nucleosome fiber.

To determine the intrinsic impact of DNA methylation (i.e., that is independent of specific methylation recognition machinery) on genome organization, we use budding yeast as a model system. *S. cerevisiae*'s genome is deprived of any cytosine methylation[30] and its structure is very well characterized[30–34]. It consists of a Rabl-like structure where the 16 centromeres are attached to the Spindle Pole Body (SPB) at one pole of the nucleus and the chromosome telomeres extend outward toward the nuclear membrane. *S. cerevisiae* does not have DNA methylation/demethylation machinery and no methylated DNA binding domain has been characterized, precluding the impact of any evolution refined biological mechanism for regulating methylation imprinting and reading. The addition of DNA methylation machinery to yeast should then yield a perfect system to check for the intrinsic impact of DNA methylation in chromatin structure.

Different authors[35,36] have shown that ectopic expression of either DNMT3a and DNMT3L or DNMT3a and DNMT1, could induce DNA methylation in yeast, but the levels of methylation achieved in both cases were too low to perform genome wide analysis of the effects of DNA methylation on chromatin structure or gene expression. More recently, Morselli et al. achieved a higher level of methylation by expressing DNMT3b at high level and collecting the cells at saturation[27]. Their results demonstrate an intrinsic anticorrelation between nucleosome and methylation and a link between DNA methylation and H3K4 and H3K36 methylation. A comparable methylation pattern was just reproduced by Finnegan et al.[37] expressing human DNMT3s in the yeast *Komagataella phaffii*.

Using a similar approach but expressing murine DNMT1, DNMT3L, DNMT3a and DNMT3b simultaneously in our *S. cerevisiae* cells, we achieved even higher methylation rates than the one reported by Morselli et al.[27], creating a perfect model for the study of the intrinsic impact of methylation in chromatin structure and function. We demonstrated that even in the absence of any directing machinery, methylations occurred in a reproducible pattern reminiscent of that seen in mammals, with methylation concentrated at linkers and depleted at nucleosomes. High levels of methylation affect gene expression in a complex manner, altering specific pathways in the cell. Quite interestingly, Hi-C experiments revealed significant changes in global chromatin structure with a noticeable increase in gene condensation. Overall, our results demonstrate that DNA methylation affects chromatin structure and function even in the absence of specific methylation-recognition machinery, suggesting that the methylation imprinting has an intrinsic impact in chromatin structure and function.

## Results

**Description of the system.** In order to reach a high level of DNA methylation, we expressed the 4 DNMTs simultaneously: yeast cells were transformed with a combination of 4 plasmids each of them expressing one murine DNA Methyl Transferase (DNMT). The de novo DNMTs, DNMT3a and 3b, and the maintenance DNMT, DNMT1, were expressed under the control of the *tet*O promoter while the expression of DNMT3L, the regulatory DNMT, was controlled by the *Gal1* promoter[38]. The expression and stability of the DNMTs were assessed by western blotting and no sign of protein degradation could be detected, even after 48 h of induction (Supplementary Fig. 1a–d). Expression of the 4 DNMTs slightly affected cell growth and cell viability (Supplementary Fig. 2a and 2b). Flow cytometry analysis performed on non-synchronized cultures after 24 h of induction showed a clear increase in the percentage of cells in G2/M in the samples expressing the DNMTs compared to the two control populations transformed with the empty plasmids suggesting that the slower growth could be due to a longer G2 phase (Supplementary Fig. 2c).

Differential gene expression analysis (Supplementary Data 1) showed that DNMTs ectopic overexpression did not induce expression of genes normally activated by stress, suggesting that the effects that we observed are not caused by stress but are a direct consequence of DNA methylation.

DNA methylation was first assessed by HPLC/MS which showed that using this approach, we could reach up to 4.2% of cytosines methylated after 38 h of induction in cells collected in stationary phase (Supplementary Table 1), and then analyzed at single base pair resolution in several independent transformants, using Illumina whole-genome bisulfite sequencing (WGBS). This was done both for cells in exponential growth phase synchronized in G1, and for cells at saturation that spent between 24 and 36 h in G1 without dividing (Supplementary Fig 2d). Methylated cytosines were, as expected, almost exclusively found in CpG context (Table 1) and showed a very reproducible pattern from one sample to another (Fig. 1a, g), a result verified using longer

**Table 1 Average methylation in CpG and non-CpG context.**

| DNMT expressed | Hours of induction | State of the culture | All contexts | | | CpG contexts | | | Non-CpG contexts | | |
|---|---|---|---|---|---|---|---|---|---|---|---|
| | | | Avg. meth | Total Cyt. | Frac. > 0 | Avg. meth | Total Cyt. | Frac. > 0 | Avg. meth | Total Cyt. | Frac. > 0 |
| DNMT1, 3a, 3 L | 30 h | Exponential - Not synchronized | 0.75% | 3,066,478 | 2.74% | 3.14% | 525,937 | 15.60% | 0.16% | 2,538,540 | 0.08% |
| DNMT1, 3b, 3 L | 30 h | Exponential - Not synchronized | 0.70% | 2,584,192 | 2.65% | 2.77% | 463,127 | 14.36% | 0.13% | 2,118,519 | 0.10% |
| DNMT3a, 3b, 3 L | 30 h | Exponential - Not synchronized | 0.49% | 2,889,913 | 2.03% | 1.97% | 502,806 | 11.46% | 0.11% | 2,385,079 | 0.04% |
| DNMT1, 3a, 3b | 30 h | Exponential - Not synchronized | 0.56% | 3,066,743 | 1.86% | 2.08% | 524,612 | 10.73% | 0.18% | 2,539,830 | 0.03% |
| None | 30 h | Exponential - Not synchronized | 0.15% | 2,732,598 | 0.00% | 0.13% | 464,513 | 0.00% | 0.16% | 2,265,439 | 0.00% |
| All 4 DNMT[a] | 27.5 h | Exponential - Not synchronized | 2.03% | 2,810,320 | 6.66% | 8.55% | 506,621 | 34.88% | 0.24% | 2,301,426 | 0.45% |
| All 4 DNMT[b] | 27.5 h | Exponential - Not synchronized | 2.13% | 2,937,375 | 6.90% | 9.14% | 522,749 | 36.43% | 0.26% | 2,412,449 | 0.51% |
| All 4 DNMT[a] | 24 h | Exponential-Synchronized in G1 | 1.75% | 4,427,610 | 6.73% | 9.72% | 676,754 | 41.0% | 0.26% | 3,750,856 | 0.54% |
| All 4 DNMT[b] | 24 h | Exponential-Synchronized in G1 | 1.51% | 4,425,531 | 6.20% | 8.36% | 676,458 | 38.1% | 0.24% | 3,749,073 | 0.44% |
| All 4 DNMT[a] | >72 h | Saturation | 5.12% | 4,425,775 | 14.7% | 27.1% | 676,923 | 67.9% | 1.11% | 3,748,852 | 5.09% |
| All 4 DNMT[b] | >72 h | Saturation | 4.75% | 4,427,957 | 13.3% | 25.2% | 677,239 | 63.4% | 1.00% | 3,750,718 | 4.22% |

Avg. meth Average of methylation in all cytosines (column 4), in cytosines only in CpG context (column 5) or in non-CpG context (column 6), Total Cyt. Total number of cytosines for which the methylation status can be determined; Frac > 0: % of cytosine with a methylation level >0
[a]Samples corresponding to replica1
[b]Samples corresponding to replica2.

reads (<10 kbp) obtained by Nanopore Technology (ONT), see Supplementary Fig. 3a–c), which allowed us to confirm the high level of methylation in long repetitive sequences in heterochromatin (Supplementary Fig. 4a, c). The global distribution of methylation across all CpG sites is shown in Supplementary Fig. 5a–c. For the cells synchronized in G1 we can see that ~50% of CpG sites have <5% methylation, but there are almost 20% of sites with methylation over 20%, and ~5% of sites with methylation over 50%. For the cells at saturation the methylation levels are generally higher, with 50% of sites having >15% methylation and 25% of sites having methylation over 50% (Supplementary Fig. 5c).

**Homogeneity of the samples**. To check the homogeneity of the samples, we examined the long nanopore derived reads, which allow assaying the methylation status at a large number of CpG sites on the same DNA molecule. We can see (Supplementary Fig. 6) that the histograms of methylation level for the 2 control samples show an almost identical distribution with a peak at around 0.02, whereas for the methylated samples from cells at saturation the peak is around 0.33. The histogram for the methylated samples from replicating cells has a peak close to the non-methylated controls, but with a long right tail.

For the methylated samples, most CpGs have an intermediate level of methylation. To investigate whether the intermediate methylation was due to heterogeneity in the population of reads (implying that the sequencing library contains a mixture of highly methylated and lowly methylated DNA molecules for a given genomic region), we modeled the distribution of methylation within reads using a mixture model with three components, one non-methylated, one lowly methylated and one highly methylated. The fit of the observed data to the mixture model was compared with that to a model with a homogenous read population using likelihood ratio tests (Supplementary Methods and Supplementary Software 1).

The results in Supplementary Table 2 show that in the samples at saturation, almost 95% of the reads are highly methylated (with roughly 30% of CpGs on the read being methylated), indicating a fairly homogenous population. On the contrary, in the samples collected in exponential phase, only 20% of the reads are highly methylated with the remaining reads having a lower methylation rate (with roughly 8% of methylated CpGs on a read being methylated). The likelihood ratio tests show high significant support for that the presence of multiple components for both samples ($p < 1.0\mathrm{e}{-16}$ in both cases). The presence of two distinct populations of reads in the samples in exponential phase suggests that the DNA methylation maintenance machinery may be not fully functional or that the replication phase in yeast is too short for the mammalian maintenance machinery, leading to a difference in average methylation between the original and daughter DNA strands.

**Effect on methylation of the different DNMTs**. We tested the functionality of each DNMT in our system by comparing the DNA methylation pattern and levels obtained when only 3 of the 4 DNMTs were expressed (Fig. 1a, b). This was performed always with replicating cells, synchronized in G1 to make the data fully comparable. For all combinations of 3 DNMTs the methylation pattern was broadly the same (Fig. 1a). The overall methylation level, however, was 2 to 3 times lower with 3 DNMTs compared to when all 4 DNMTs were expressed, demonstrating that each DNMT is functional (Fig. 1b).

While the different DNMT combinations displayed broadly the same pattern of methylated CpGs, when examined in detail some differences emerged. To check whether these differences are due

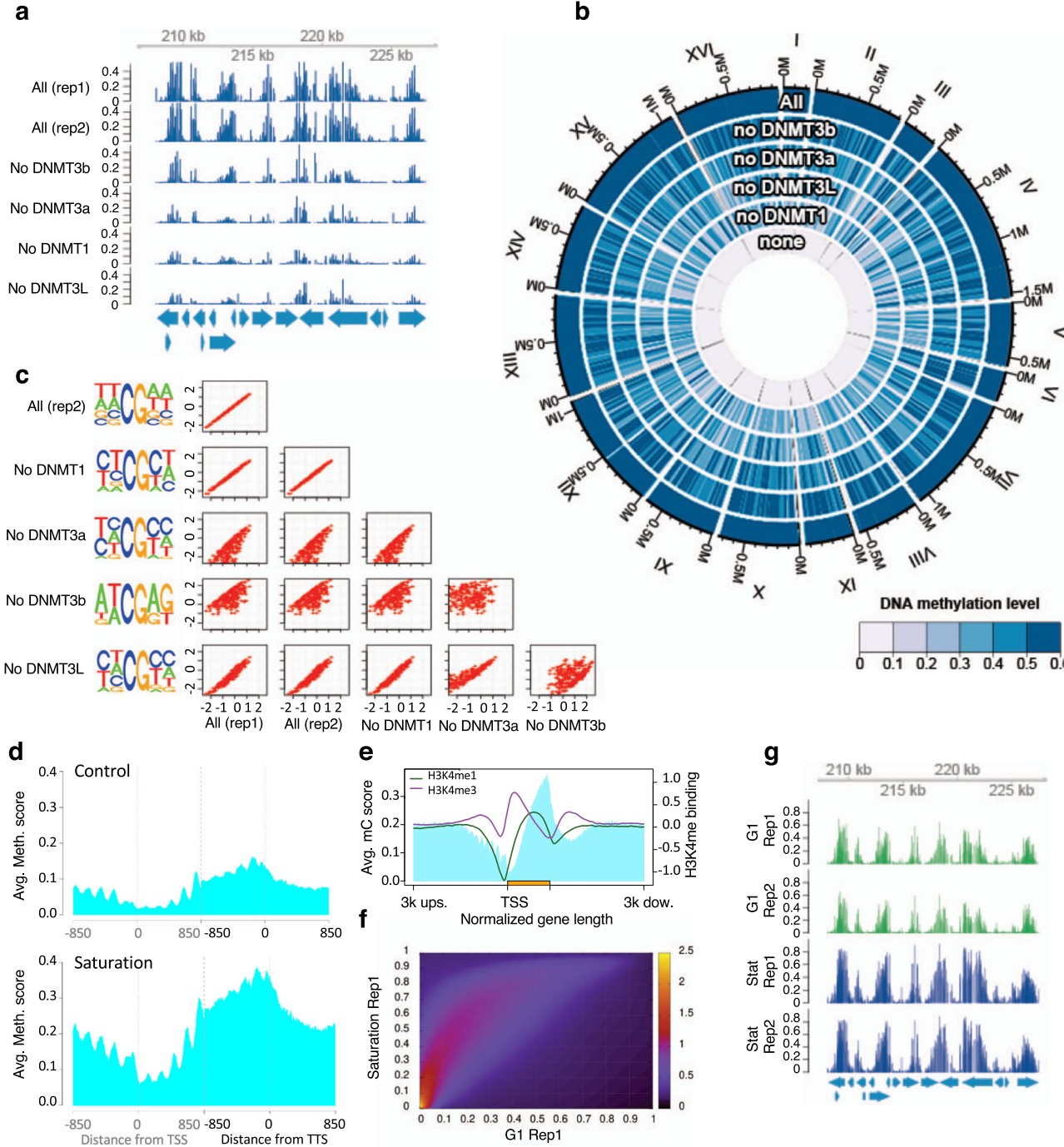

**Fig. 1 Methylation pattern across several samples and along the gene body. a** The pattern of methylation is conserved in all samples as illustrated for this 20 kb region of chromosome III (208,135..227,458) where the level of methylation at each position is represented for two samples with the four DNMTs expressed and four samples with each combination of 3 DNMTs. All the samples come from replicating cells, synchronized in G1. **b** Circular plot comparing DNA methylation for the samples in **a** and a control without methylation (None). Methylation levels decrease when one DNMT is missing, the strongest effect being without DNMT1 and the mildest effect when DNMT3b is not present. **c** Correlation between the sequence effect on methylation status among samples with different combinations of three DNMTs. Motif effects are estimated from logistic regression (for details see Methods) and correlation plots are produced for each pair of samples. Motifs have nearly the same effect in the two replicates with all DNMTs induced (correlation coefficient is cor = 0.999) and when DNMT1 (cor = 0.996) or DNMT3L (cor = 0.954) are removed. In contrast, the estimated effect of some motifs on methylation probability is different in samples lacking DNMT3a (cor = 0.793) or DNMT3b (cor = 0.631). The left panels show the sequence logo of the motifs preferentially methylated in each sample. **d, f, g** Comparison of methylation pattern in samples in exponential phase and at saturation. **d** Average methylation level around TSS and TTS (850 bp upstream and downstream from each point). **e** Superposition of the pattern of DNA methylation and the pattern of H3K4 methylation along the average gene body (from 3 kb upstream TSS to 3 kb downstream TTS). DNA methylation preferentially occurs where H3K4 is not methylated. **f** Heatmap showing the correlation between methylation probabilities in samples in G1 and at saturation. **g** The pattern of methylation is conserved in samples in G1 and samples at saturation as illustrated for this 20 Kb region of chromosome III (208,135..227,458) where the level of methylation at each position is represented for 2 replicas of each condition, G1 (in green) and saturation (in blue).

to intrinsic sequence specificity of the different DNMTs, logistic regression was used to assess the effect of local sequence context on the methylation rate at CpG sites. This analysis was applied to 6 samples (2 samples with all 4 DNMTs to assess the reproducibility of the results, and the 4 samples each lacking one of the DNMTs). Figure 1C shows that the two replicates with all 4 DNMTs being expressed provide very similar results, proving the robustness of the experiments. Removing DNMT1 or DNMT3L does not significantly alter the sequence preferences found in the experiments with the 4 DNMTs active. In contrast, removing either DNMT3a or DNMT3b has a large effect on the sequence context. In general, cells lacking DNMT3b show a strong bias for CpGs in the 5′ ATCGAG 3′ motif (with the percentage of methylated CpGs in an ATCGAG motif being more than six times lower when DNMT3b is removed, compared to the samples with one of the other DNMTs missing or to the sample with all 4 DNMTs induced). Cells lacking DNMT3a also show a sequence bias, in this case towards sequences that are more C rich. Note that the lack of methyl DNA binding protein makes these sequence preferences intrinsic of DNMTs and not the result of positioning of the methylases in certain sequences due to auxiliary proteins.

**Pattern of methylation.** Despite the lack of specific proteins directing epigenetic signaling of DNA, the pattern of methylation obtained in our model organism is very similar to that observed in higher eukaryotes, with DNA methylation being low at the Transcription Start Site (TSS) increasing toward the end of the genes and reaching a maximum at the Transcription Termination Site (TTS) (Fig. 1d). While we cannot rule out this pattern being due to selection (i.e., that cells with no methylation at promoters have a selective advantage), the fact that the pattern is established very early on (6 h after the induction of the DNMTs) and remains stable across all time points indicates that this has a minor effect, if any.

Previous studies have linked DNA methylation with histone post translational modifications, DNMT3L being recruited by unmethylated H3K4 and DNMT3A and 3B recruited by H3K36me2 and H3K36me3, respectively[27,39–42]. We confirmed that the H3K4 methylation pattern in the stationary cells both in the control and in the methylated sample was the same as the one described in exponential cells, and that DNA methylation was anti-correlated with H3K4 trimethylation (Fig. 1e), suggesting that H3K4 and H3K36 methylation are capable of tightly controlling DNA methylation even upon DNMTs overexpression by a direct mechanism.

**DNA methylation and nucleosome positioning.** To investigate the correlation between DNA methylation and nucleosome positioning, we obtained MNase-Seq data of the samples with all 4 DNMTs or with empty plasmids in stationary phase. We observed that DNA methylation was anti-correlated with nucleosomes and tended to be accumulated in the linker regions (Fig. 2a, c). These results agree with the bulk of in vivo[25–28], in vitro and in silico data[18,19] and ruled out the suggestion that methylation is favored at well-positioned nucleosomes[22,23].

We observed a small (4%), but very significant increase (*p* value 2.2e−16) in the number of fuzzy nucleosomes and a mirror decrease in the number of well-positioned nucleosomes in the methylated samples (Table 2). To test if this effect was due to DNA methylation and not to the presence of the DNA methyltransferases themselves, catalytically inactive mutants of the DNMTs (DNMT3b[P656V/C657D], DNMT3a[P705V/C706D] [43] and DNMT1[C1229S] [44]), were expressed in yeast and MNase-seq was performed. PCA (Supplementary Fig. 7a) and NucleR analysis

(Supplementary Fig. 7b) confirm that the effect we observed was mostly due to DNA methylation.

In well-positioned nucleosomes, DNA methylation was almost absent at the dyad and increased toward the entry and exit points of the nucleosome, while fuzzy nucleosomes had higher methylation levels with a more constant level across the nucleosome (compare Fig. 2c with Fig. 2c). Detailed analysis (Fig. 2e), failed to detect any periodicity in the methylation within the nucleosome bound sequence, arguing against a preferential methylation of DNA at accessible sites of the nucleosome. Considering that all methylases were active, this suggests that (at least in the absence of directing machinery) all DNA segments covered by histones are equally inaccessible to methylation imprinting.

Figure 3a shows some representative nucleosome fiber structures for the DUG2 gene (chromosome II) as a test case, modeled from MNase-seq signals for control and methylated cells. Both ensembles of structures reproduce the experimental results to a great extent (Fig. 3b, c and Supplementary Fig. 8a, b) and confirm the presence of fuzzier nucleosomes in methylated cells. Due to the presence of fuzzier nucleosomes, a wide range of nucleosome configurations are possible leading to the sampling of very elongated conformations with large radius of gyration (see Fig. 3a, lower panel, and Supplementary Fig. 8c). The diversity of nucleosome arrangements was further analyzed by comparing the 3D distances between $N$ and $N + x$ nucleosomes (Supplementary Fig. 8d), which in methylated cells show a more dispersed profile. Finally, it is noticed that regions with high degree of methylation probability are mainly linker DNA (Fig. 3b, c and Supplementary Fig. 8b), though the probability of finding some nucleosomal DNA methylated is not insignificant, as shown in some fiber configurations (Fig. 3a, lower panel). This is because those nucleosome positions are retained despite having low coverage in the MNase-seq signal.

Analysis by Nucleosome Dynamics[45] of the crucial region around the promoter demonstrates that high level of methylation induces more dynamic and fuzzy nucleosomes (Fig. 4a), as well as a statistically significant ($p < 2.2e{-}16$) narrowing of Nucleosome Free Regions (NFRs; Fig. 4b, c). These changes, which are typically considered to be signals of gene inactivation[46,47] cannot be explained here by the coordinated effect of methyl-DNA binding proteins coupled to chromatin remodelers and must be then considered intrinsic to the changes on physical properties of DNA induced by methylation and its direct impact in protein-DNA interaction. The fact that high methylation is correlated also with narrower NFR also at the 3′ end of the genes (Supplementary Fig. 9) agrees with this hypothesis.

**DNA methylation and gene expression.** Despite the lack of any directing mechanism, the methylation pattern is quite homogenous throughout the gene only in lowly expressed genes while in the highly expressed ones, DNA methylation is low at the promoter and increases toward the end of the gene, suggesting a direct link between DNA methylation and gene expression (Fig. 5a) in the absence of any specific methylation-recognition mechanism. A differential expression analysis (Fig. 5a) shows that genes which are very lowly methylated do not change their expression level between the control and transformed cells, while high methylation levels lead to important changes in gene activity in both directions: towards greater and lower expression (Fig. 5c and Supplementary Data 1). Although we cannot discard any indirect effect caused by the binding of the DNMTs on the chromatin, we see a very strong correlation between gene expression and methylation level for a subset of genes involved in meiosis and that appear to share a common sequence in their

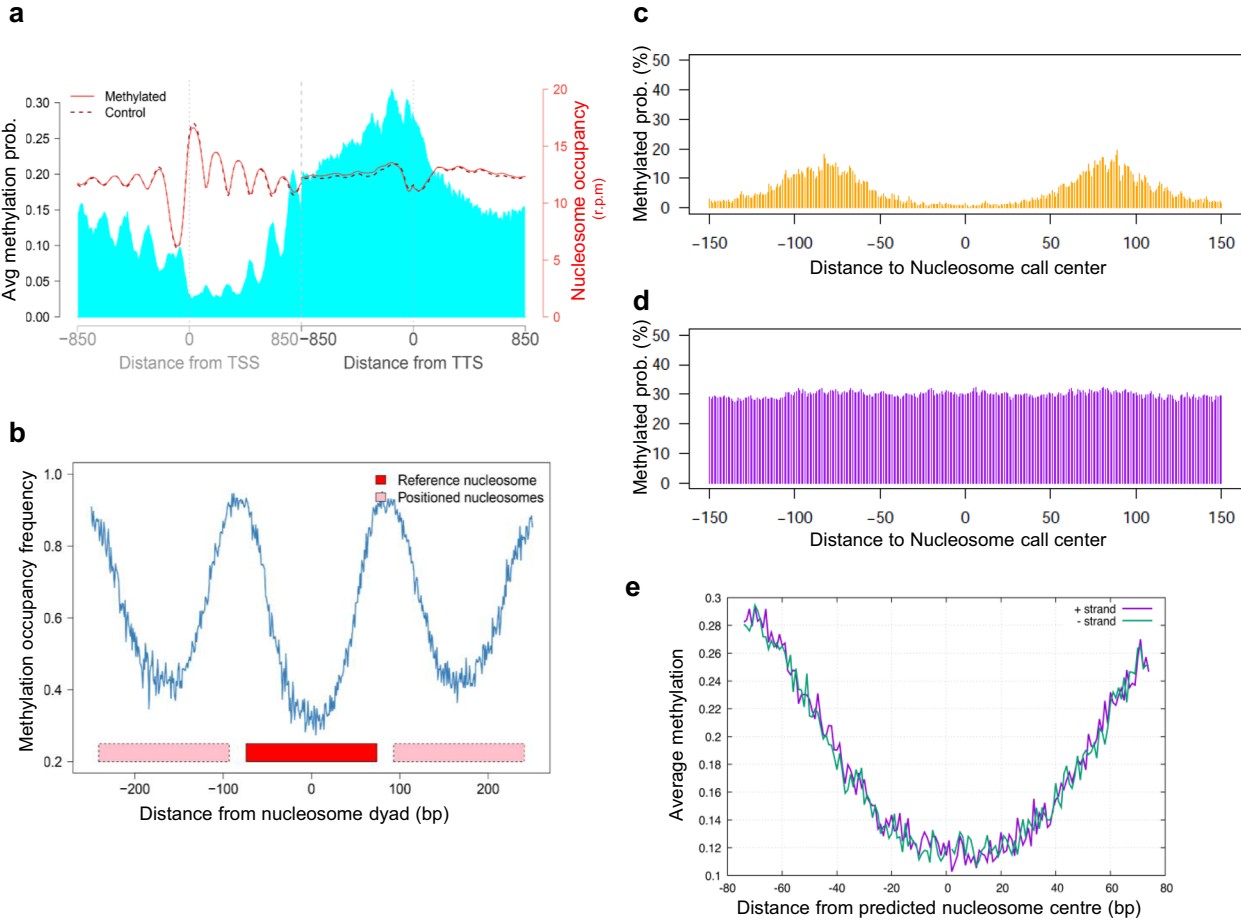

**Fig. 2 Correlation between DNA methylation and nucleosome coverage genome wide. a** Nucleosome positioning (in red) in a sample before (dashed lane) and after (plain lane) induction of methylation and average methylation probabilities (in blue). Plots are built around TSS and TTS (850 bp upstream and downstream from each point). Average nucleosome positioning does not change drastically upon methylation. **b** Percentage of CpG with methylation probability above 0.01 around well-positioned (W) nucleosomes. Nucleosome calls were considered well-positioned when nucleR peak width score and height score were higher than 0.6 and 0.4, respectively. DNA methylation is anti-correlated with nucleosome occupancy in W nucleosomes. Average methylation probability around nucleosome call center (150 bp upstream and downstream) for (**c**) Well positioned and (**d**) Fuzzy nucleosomes. **e** Average methylation probability per strand around nucleosome call center (75 bp upstream and downstream) of well-positioned nucleosomes.

**Table 2 Distribution of the nucleosomes classified by NucleR.**

| Sample | Fuzzy nucleosomes | Well positioned nucleosomes | Not determined | Total |
|---|---|---|---|---|
| Ctrl Rep1 | 34,759 (47.36%) | 38,320 (52.22%) | 307 (0.42%) | 73,386 |
| Ctrl Rep2 | 36,322 (49.64%) | 36,544 (49.95%) | 298 (0.41%) | 73,164 |
| Met Rep1 | 39,400 (53.12%) | 34,362 (46.33%) | 403 (0.54%) | 74,165 |
| Met Rep2 | 38,222 (51.61%) | 35,425 (47.83%) | 416 (0.56%) | 74,063 |

regulatory region (Fig. 5c, d), corresponding to the binding site of Ume6p, a subunit of the histone deacetylase complex Rpd3p known to repress early meiotic gene expression. It is tempting to hypothesize that methylation of a Ume6p binding site known as URS1, could affect intrinsically Ume6p binding directly (through changes in direct protein-DNA interactions or indirectly through changes in chromatin structure), leading to a deregulation of its target genes. Supporting this hypothesis, we observed that the level of expression of the target genes increases proportionally with the level of methylation of the Ume6p binding site (Supplementary Fig. 10a and Supplementary Table 3). This was further confirmed in vitro by showing that Ume6p has more affinity for an unmethylated URS1 site than a methylated one (Supplementary Fig 10b, c). Finally, for most of Ume6p target genes, we

observe a 5–10 bp shift of the −1 or +1 nucleosome (Supplementary Fig. 11), consistent with changes in expression. In summary, it seems that two physically driven mechanisms: changes in protein-DNA binding interactions due to the presence of a methyl group and methylation-induced nucleosome rearrangements work in a coordinated way to induce a change in gene activity which would be typically assigned to the effect of specific methyl DNA binding proteins, which are absent in *S. cerevisiae*.

**DNA methylation and genome 3D structure.** We performed Hi-C experiments in control and methylated populations at saturation in two replicas processed in parallel to explore the intrinsic effect of DNA methylation in the global chromatin structure

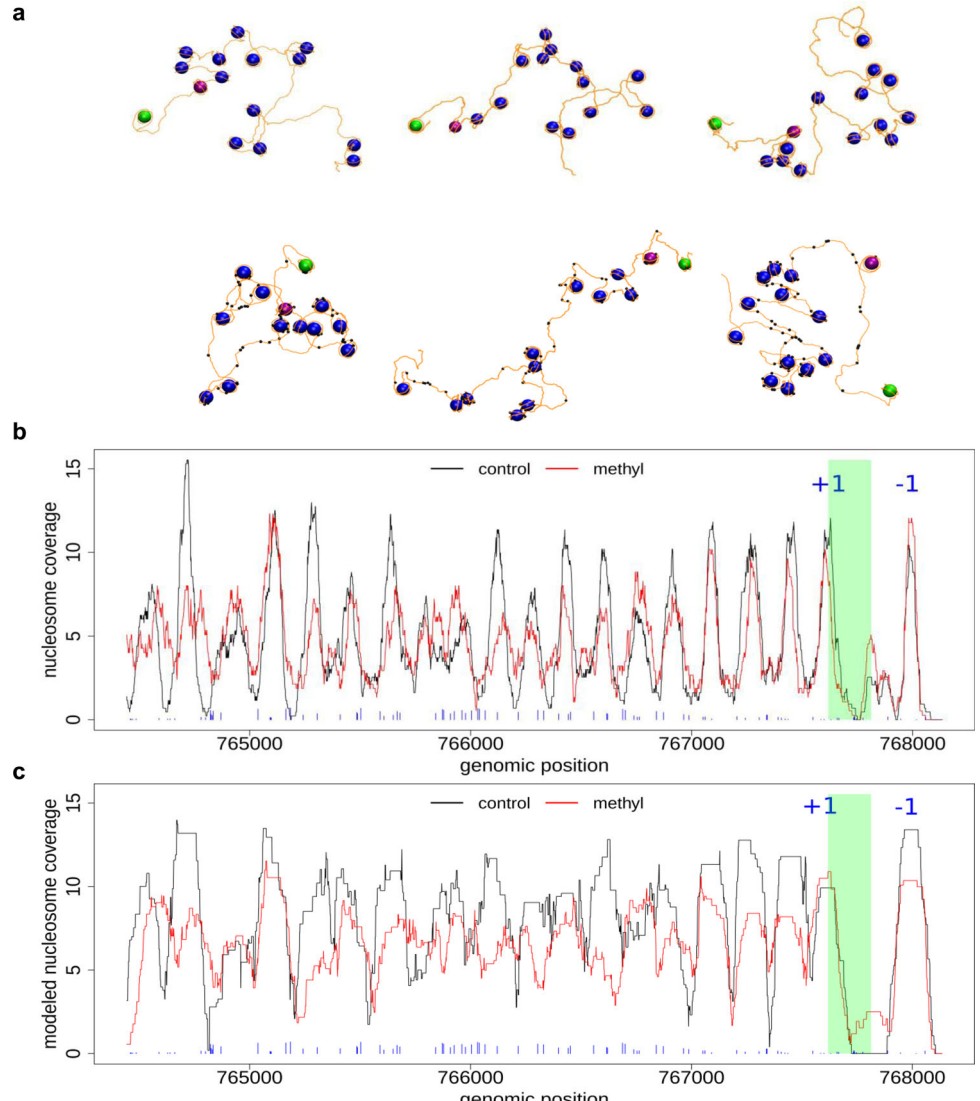

**Fig. 3 Modeled nucleosome fibers from MNase-seq data. a** Modeled nucleosome fiber for (top panel) control and (lower panel) methylated cells. DNA base pairs are depicted as orange lines and nucleosomes as blue spheres (+1 and −1 nucleosomes are depicted in purple and green, respectively). Methylated regions (signal above 10% coverage) are highlighted as black balls. Experimental (**b**) and modeled (**c**) nucleosome position coverage for DUG2 gene (chromosome II). The TSS is highlighted in green and nucleosomes −1 and +1 are indicated. The blue sticks in the bottom correspond to methylated regions with their height proportional to methylation extent.

(Supplementary Fig. 12a–d). As shown in Fig. 6a, b (and Supplementary Fig. 13a, b), DNA methylation leads globally to an increase in *cis* (especially close to the centromeres (<100 kb); Fig. 6c, d) and a very significant decrease in *trans* contacts (Fig. 6c, e). To obtain further insights into the effect of DNA methylation on chromatin structure, we modeled the spatial organization of each chromosome using a restraint-based model derived from the interaction counts (see Methods) from which we obtained ensembles reproducing with an astonishing quality the HiC maps (Supplementary Fig. 14). Clearly, in all chromosomes (except the short chrI), the 100 kb region centered around their centromeres is more condensed upon DNA methylation (Fig. 6f), decreasing the overall chromosome flexibility with the effect being very clear for chromosomes V, IX, XI and XV (Fig. 6g and Supplementary Fig. 14). Very interestingly, the nucleosome depletion that is observed at the centromeres in stationary phase in the control sample (compared to cells in exponential growth) is not detected in the methylated one (Supplementary Fig. 15), suggesting that the profound reorganization of the centromeric

region occurring in quiescent cells is reduced in the methylated samples.

The three-dimensional arrangement of telomeres is also largely modified by methylation, as we observed a significant decrease in the number of interactions (Supplementary Fig. 16), resulting in a large dispersion of telomeres in methylated cells, something that is visible in the HiC-derived ensembles (Fig. 6h, i). It is worth noting that telomeres tend to cluster in quiescent cells to form hyperclusters[48] but it seems that this reorganization is not happening when DNA is methylated. Once again, intrinsic changes linked to methylation lead to a 3D organization that is closer to the one described in exponential growth (which is when the expression of the DNMTs started to be induced), than the one we and others observed in quiescent cells.

Comparing the changes in interactions between the control and methylated samples for each individual chromosome (Supplementary Fig. 17), we observed the strongest effects of methylation for chrIII (largest increase in intra-chromosomic-contacts, Fig. 7a–e) and for chrXII (largest decrease in intra-

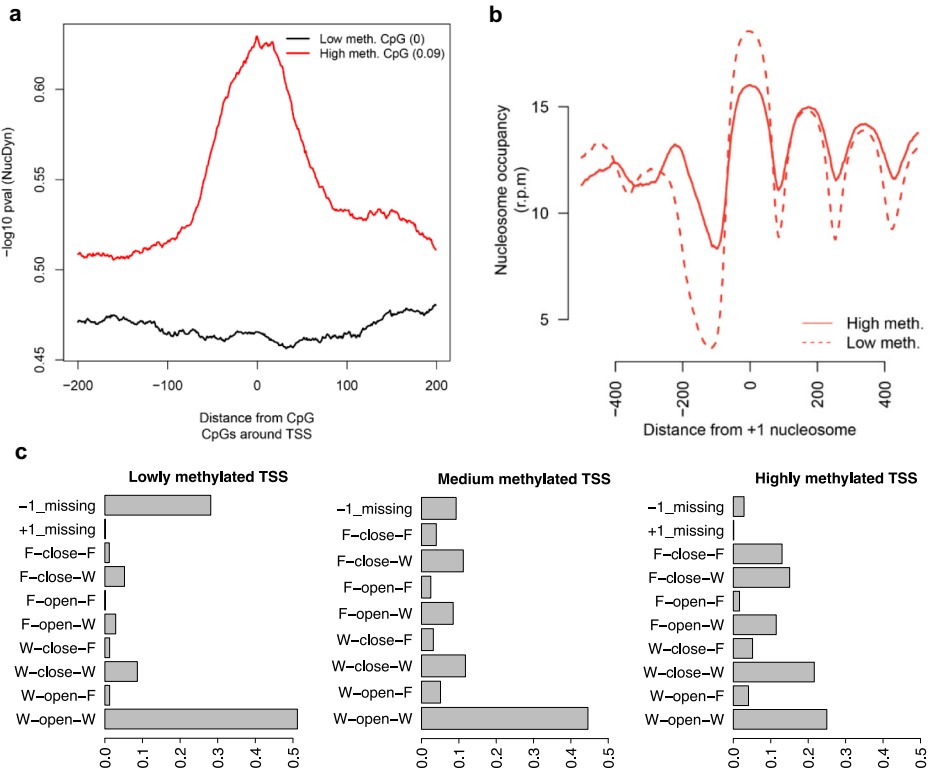

**Fig. 4 Nucleosome dynamics upon methylation at promoters. a** NucDyn score around highly methylated (in red) and lowly methylated CpGs (in black) at promoters. **b** Nucleosome coverage around +1 nucleosome for genes with highest (top 10%, plain red lane), lowest (bottom 10%, dotted red lane) or medium (45–55%, doted black lane) level of CpG methylation around promoters. **c** Nucleosome architecture around promoters according to their methylation level.

chromosomic contacts, Fig. 7f–h). Looking closer at chrIII, we noticed an important gain of interactions in the methylated sample between the left telomeric region containing the silenced HMLα and the peri-centromeric region (Fig. 7a–c). This gain of interaction correlates with a significant decrease of the distance between the HMLα and the MATa loci (Fig. 7d) which leads to very significant changes in the ensemble of chromosome conformations compatible with the HiC restrains (Fig. 7e). This result was validated by live-cell microscopy using the system described in[49,50] where we can see that the distance between HML and MAT is significantly smaller in the methylated cells compared to the control while the distance between HML and HMR and between MAT and HMR are conserved (Supplementary Fig. 18).

ChrXII is a very peculiar chromosome, as it carries the rDNA locus that consists of 150–200 repeats of the rDNA genes and localizes at the nucleolus where ribosomal RNAs are transcribed. In the control sample, interactions between the upstream and downstream regions are found, which can be explained by the global decrease of transcription occurring in stationary phase leading to a more compact nucleolus compared with exponential growth[51]. However, in methylation conditions (Fig. 7f–h), the segregation between the upstream and downstream regions of the chromosome is very strong, reproducing again the situation expected for a cell in exponential growth. The absence of interactions between the two domains probably allows a general increase in the relative flexibility that could explain why the general reduction in flexibility related to methylation seen in other chromosomes is inverted for this chromosome (Fig. 6g).

Globally, the control samples show a quite spherical nucleus (Fig. 8a) with the chromatin concentrated in the exterior (Fig. 8b), while in the methylated samples the nucleus tends to elongate with

a more dense packing of chromatin in the interior (Fig. 8d, e). In both cases, centromeres are all localized to one pole of the structure, organized as a rosette (Fig. 8c, f), but they do not appear as clustered as reported for the interphasic nucleus[33,34,52].

## Discussion

Yeast expressing murine methyl transferases show, in vivo, a specific, reproducible, pattern of DNA methylation that is very similar to that of genes-containing regions in mammals. This is a striking result as yeast is an organism deprived of any DNA methylation machinery and is not ready to place or recognize methylation marks. Methylation leads to a slight decrease in the viability and in the doubling time maybe caused by a longer than normal G2/M phase. However, those changes are moderate, and again it is quite surprising that an organism not prepared to have methylated DNA tolerates well a large amount of methylation in its genome.

Our synthetic model system helped us to highlight some previously unknown intrinsic sequence specificity for the methyl-transferases, i.e., those that cannot be explained by accessory proteins. For example, little sequence specificity is found for DNMT1 and DNMT3L, while significant sequence specificities are found for DNMT3a and DNMT3b. Thus, cells lacking DNMT3b show a strong bias for methylation of CpG in 5' ATCGAG 3' motif, while cells lacking DNMT3a seems to have a bias toward CpG sequences embedded in C-rich environments. Again, these differences are intrinsic and not coupled to any specific directing mechanism, which suggests a certain level of specificity of the DNMTs.

Methylation in our engineered yeast model is preferentially located between nucleosomes, and when it occurs in nucleosome-

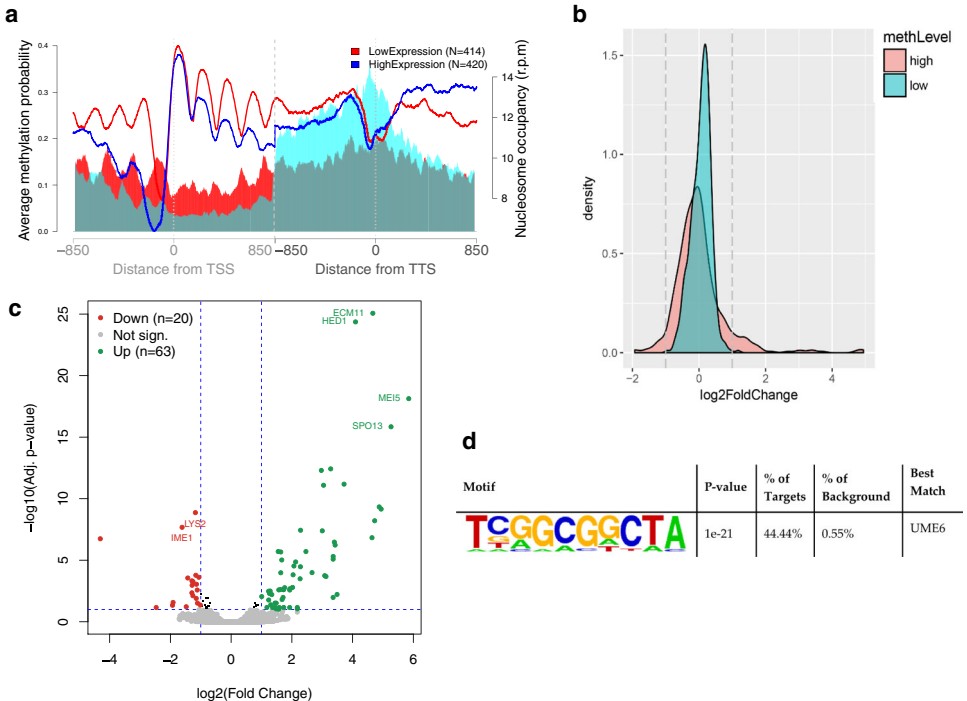

**Fig. 5 Correlation of DNA methylation, nucleosome positioning and gene expression. a** Methylation probability (shaded area) and nucleosome coverage (solid lines) around genes with expression in the top 10% (blue) or bottom 10% (red) expression level. **b** Log2 of the fold change in expression for genes with highly and lowly methylated promoters. **c** Gene expression difference between methylated and not methylated samples. RNA level difference is plotted on the x-axis and the Adj. *p* value on the y-axis. Downregulated (20 genes) and upregulated (63 genes) genes are shown in red and green, respectively. The genes with the highest changes are highlighted. **d** Promoter motif enrichment for genes with highly methylated promoters and increase in expression level using HOMER which uses a cumulative hypergeometric distribution to score motif enrichment in the target set compared to the background set. The p-value is not corrected for multiple testing.

occupied regions, it is associated with significant alterations in nucleosome positioning reflected in an increase in nucleosome fuzziness. The fact that methylated DNA is less frequent at nucleosomes confirms our previous in silico and in vitro models[18,19] but does not rule out the possibility that in higher eukaryotes methylated-DNA binding domains might stabilize the presence of methylated CpG in mammal nucleosomes, leading to a situation of "loading-spring" which might facilitate fast nucleosome reorganization upon release of the stabilizing protein. There is, however, no question that methylation and nucleosome position are intrinsically anti-correlated, confirming previous results in mammalian cells from NOME-seq[29] where the authors observed that DNA methylation and nucleosome occupancy were strongly anti-correlated surrounding CTCF sites. Furthermore, our results demonstrate that there is not any short-distance periodicity pattern which might indicate methylation at periodically exposed regions of nucleosome DNA. Very interestingly, when methylation occurs in the promoter region it tends to narrow the NFR region, a fingerprint of lowly expressed genes in mammals, which is found here in absence of any methylation-specific chromatin remodeler.

In mammalian cells, the relationship between methylation and gene expression is complex, with high levels of gene expression often associated with low promoter methylation but elevated gene body methylation, and the causality relationships are not clear. Our simple synthetic system allows a more direct interrogation to the relationship between methylation and gene expression. We observed that despite the lack of specific methylated DNA binding domains, highly and lowly expressed genes have quite different methylation profiles, with much higher levels of methylation near (±850 bp) the TSS of silent genes while very

actively expressed genes have much higher methylation levels at the TTS. In the absence of specific proteins modulating this profound difference, we can speculate that nucleosome positioning is one of the main factors responsible for this differential behavior, which suggests that methylation and nucleosome positioning might act in concert in the regulation of gene function in mammals, adding an extra layer of control of gene expression.

Our results suggest that methylation can also directly affect the binding of a transcription factor highlighting another intrinsic role of methylation in gene activity, which is often ignored, but that can be important in modulating DNA-protein binding free energy. Interestingly, we found that the methylation-induced change in protein-DNA binding is responsible for a dramatic increase of expression of early meiotic genes. In that case, our results suggest that the methylation affects the intrinsic binding of the histone deacetylase complex Rpd3p, thus hindering the placement of repressive marks on these genes.

Analysis of HiC data shows that the characteristic Rabl configuration previously described[30,33,34] is maintained in stationary phase both in the control and in the methylated sample. However, methylation induces an increase of intra-chromosomic contacts and a significant decrease of the inter-chromosomic ones, which leads to significant changes in the chromosome conformation. We observed that a significant (Fisher's exact test, $p = 1.76e{-}7$) proportion of the interactions that are gained or lost upon methylation involves regions containing one or several tRNA genes, supporting the involvement of RNA polymerase III in the overall organization of chromatin structure[53].

Changes in heterochromatin regions of *S. cerevisiae* (telomeres, the mating type locus and the rDNA locus) are also clear upon

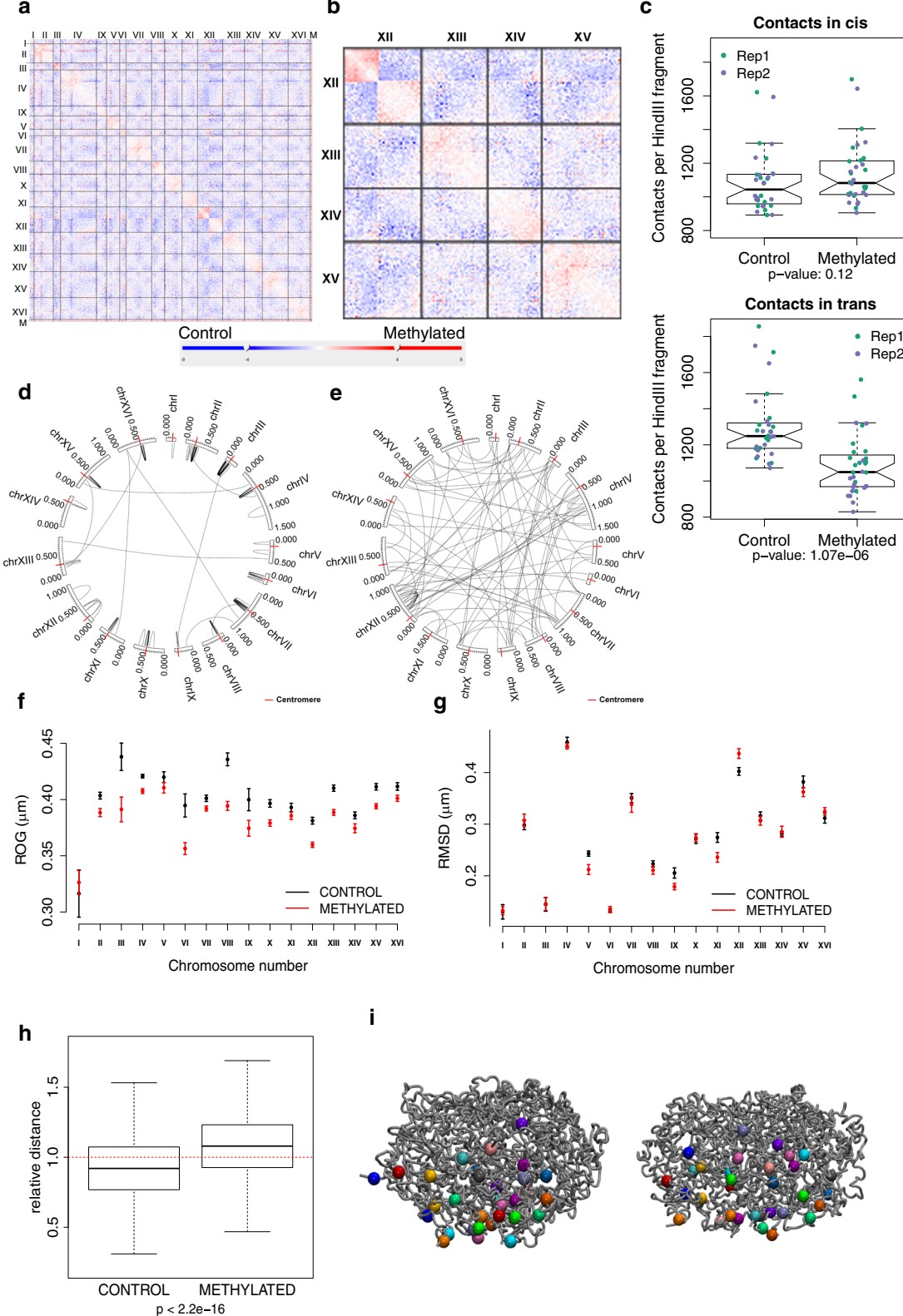

methylation, mimicking the situation found in mammals. First, we observed a general loss of interactions between telomeres which leads to a significant change in chromosome structure and that can be explained by a generally higher rigidity of the chromatin upon methylation. Second, we observed that chromosome III underwent some conformational changes in response to

methylation, with the silenced HMLα locus getting closer to the MAT locus, a conformation expected in exponentially growing MATa cells[49,54], but not in stationary cells. Finally, the separation commonly observed in yeast between the two regions of chrXII delimited by the rDNA locus is clearly weaker in our cells in stationary phase when compared to the structure published for

**Fig. 6 Effect of DNA methylation on 3D genome structure.** Differential contact frequencies in control and methylation-induced samples in replica 1 for (**a**) whole genome and (**b**) four chromosomes. Blue indicates interactions with a higher frequency in the non-methylated control samples compared to the methylated samples, while red indicates the converse. **c** Comparison of contact frequencies between control and methylated Hi-C samples in *cis* (±50 Kb from the centromere, top panel) and in *trans* (lower panel). Contact frequencies for $n = 16$ chromosomes on two biological replicas for each of the two conditions. Boxplot lower and upper hinges correspond to the first and third quartiles, respectively, and the middle line represents the median. The upper and lower whiskers extend from the hinge to the largest and smallest values, respectively, but no further than 1.5 × IQR (Inter-quartile range) from the hinge. Notches extend to ±1.58 IQR/sqrt(n) and give a rough 95% confidence interval for the median. Circos plots displaying the significant (FDR < 0.5) differential interactions identified with diffHiC: **d** gained interactions (log2FC > 1) are clustered around the centromeres (red tick marks) and **e** lost interactions (log2FC < −1) preferentially occur between chromosomes. Structural changes measured on the ensemble of structures obtained with our restraint-based 3D model for each chromosome: **f** Mean radius of gyration computed around centromeres (±100 kb) and **g** flexibility of each chromosome measured by the RMSD of bead positions for the control (black) and methylated (red) samples. Values ± standard error are plotted ($N = 260$ and 482 ensemble structures for control and methylated samples, respectively). **h** Relative distance (distance divided by mean) between all telomeres in the ensemble of 3D structures ($n = 496,000$ distances). Outliers were not plotted (min/median/max values are 0.081/0.920/1.901 for control and 0.099/1.080/1.919 for methylated conditions). The *p* value is from a two-sided *t*-test for the difference in means. **i** Whole-genome 3D model for a representative structure from the ensemble for the control (Left) and methylated sample (Right). All chromosomes are represented as gray tubes, and the telomeres represented as colored spheres with a different color for each chromosome.

cells in exponential phase[30], but the situation is reverted in methylated samples (in stationary phase), suggesting that methylation of the rDNA can freeze the heterochromatin in exponential phase-like conformation.

Our previous results[18,19] strongly suggest that in general, methylation increases the stiffness of DNA and this should impact the structure of the chromatin both locally and globally. This is confirmed here by Hi-C experiments coupled to 3D modeling that show a condensation of the centromeric region for each individual chromosome, resulting in more rigid and condensed chromosomes unable to transiently interact with other chromosomes. Locally, at the gene level, the situation is more variable, as the higher rigidity of methylated DNA leads to fuzzier nucleosome architectures and longer linkers (as shown by MNase-seq experiments), which in turn can lead to a local increase in the structural variability of the nucleosome fiber, dominated by the length of the linkers. Overall these results demonstrate that the local increase in rigidity induced by methylation have direct consequences on the global structure of chromatin, even in the absence of protein machinery ready to recognize methylation signals.

Taken together, our results suggest that in the absence of any mechanism directing methylation and reading the methylation signals, methylation leads to significant changes in chromatin structure and function. Very interestingly and quite surprisingly, many of these methylation effects resemble those annotated to methylation in mammals, where an exquisite machinery for imprinting and reading of methylation signals is present. It can be concluded then that changes in physical properties of DNA induced by methylation (reflected in alterations in DNA deformability or in DNA-protein interactions) can be responsible for a significant part of the phenotypic effects triggered by DNA methylation in mammals and that the cellular machinery involved in DNA methylation just amplifies an intrinsic signal coded in the physical properties of DNA.

Changes in methylation across the genome are key features of developmental pathways in both normal differentiation[55,56] and many cancers[2]. We observed that in our synthetic model, methylation makes the 3D genomic structure in the stationary phase closer to that of G1 cells than in the unmethylated samples, which suggest that one of the intrinsic roles of DNA methylation could be to freeze chromatin conformation to maintain the state of cell.

## Methods

**Plasmid construction**. pYADE4 yeast plasmids encoding full length *DNMT1* and *DNMT3a* with modified sequences around the translation start sites were kindly provided by Dr Jan Fronck, pYES3/CT encoding *DNMT3b* was provided by Dr Shen Li[35,57]. *DNMT3L* cloned into pYES3/CT to produce a Nterminal FLAG tagged DNMT3L was provided by Dr Jia-Lei Hu[36].

pCM188 (marker cgURA3) and pCM185 (marker cgHIS or cgLEU), centromeric vectors[38] which differ for the number of Tet operators (respectively 2 and 7) were kindly provided by Dr Jessie Colin.

*Sma*I restriction fragment (from pYADE4-DNMT1) containing full length *DNMT1* cDNA was inserted at *Pme*I site of pCM185 (LEU) to give pCM185(LEU)-DNMT1.

*Bam*HI-*Mlu*I restriction fragment (from pYADE4-DNMT3a) containing full length cDNA of *DNMT3a* was ligated to pCM185 (HIS) linearized with *Bam*HI and *Mlu*I to give pCM185(HIS)-DNMT3a.

*Bam*HI-*Not*I restriction fragment from pYES3/CT-DNMT3b containing full length *DNMT3b* was ligated to pCM188 (URA) linearized with *Bam*HI and *Not*I to obtain pCM188 (URA)-DNMT3b.

*DNMT mutants' constructs* – Plasmids encoding the catalytic mutants DNMT3b[P545V/C657D], DNMT3a[P705V/C706D] and DNMT1[C1220S] were constructed using the Q5® Site-Directed Mutagenesis Kit. The mutagenesis was performed on the pCM188-DNMT3b, pCM185-DNMT3a and pCM185-DNMT1 plasmids. Sequences of oligonucleotides for mutagenesis, subcloning or gel retardation assay are listed in Supplementary Table 4.

*Live cell imaging constructs* – DNMT3a-DNMT3b-pBEVYGU: The full length DNMT3b coding sequence was amplified by PCR on pCM185-DNMT3b plasmid using the pcrDNMT3b_Kpn1 5´_F and pcrDNMT3b_EcoRI 3´_R oligonucleotides. The PCR product was digested with *Kpn*I and *Eco*RI and ligated to the pBEVY-GU (URA) plasmid. *DNMT3a* gene was obtained by digestion of the pCM185-DNMT3a plasmid with *Bam*HI and *Xba*I. The restriction fragment was ligated into the pBEVY-GU-DNM3b vector that had been previously opened with the same restriction enzymes.

DNMT3L-DNMT1-pBEVYGT: The *DNMT3L* gene was amplified by PCR on the pYES-DNMT3L plasmid using the pcrDNMT3L_SacI1_5´_F and pcrDNMT3L_EcoRI 3´_R oligonucleotides. The PCR product was digested with *Sac*I and *Eco*RI and ligated to the pBEVY-GT (TRP) plasmid. The *DNMT1* gene was obtained by digestion of the pYADE4-DNMT1 with *Spe*I and *Xba*I. The restriction fragment was ligated into the pBEVY-GT-DNM3L vector that had been previously opened with the same restriction enzymes.

*Recombinant Ume6p construct* – UME6-pET14b: The entire coding sequence of *UME6* was amplified by PCR from yeast genomic DNA using the UME6-F and the UME6-R oligonucleotides. PCR product was digested with *Nde*I and *Xho*I. The restriction fragment was ligated into the pET14b vector previously opened with the same restriction enzymes. This plasmid adds a 6xHis tag at the N terminal part of the sequence.

**Yeast strains and culture conditions**. Strain YPH499 (*Mata ura3-52 lys2-801 ade2-101 trp1-Δ63 his3-Δ200 leu2-Δ1*) was transformed with 2, 3 or 4 expression plasmids by the standard lithium acetate procedure. Transformants were selected on plates of appropriate selective medium with 2% Raffinose and 10 µg/ml doxycycline to repress any expression.

Selected transformants (2 to 4 transformants per combination of plasmids) were grown on selective liquid medium with 2% Raffinose and 10 µg/ml doxycycline up to OD600 = 0.5. Then, yeast cells were spun 10 min at $1000 \times g$, washed twice with sterilized water, and resuspended into selective media with 1% Raffinose and 2% Galactose without doxycycline to allow expression of all four DNMTs. For experiments on synchronized cells, cells were treated with alpha-factor (3 µM final) for 4 h to synchronize cells in G1 or with Nocodazole to synchronize cells in G2.

After different times of induction, cells were collected and treated for subsequent experiments: protein extraction for western blotting, gDNA extraction

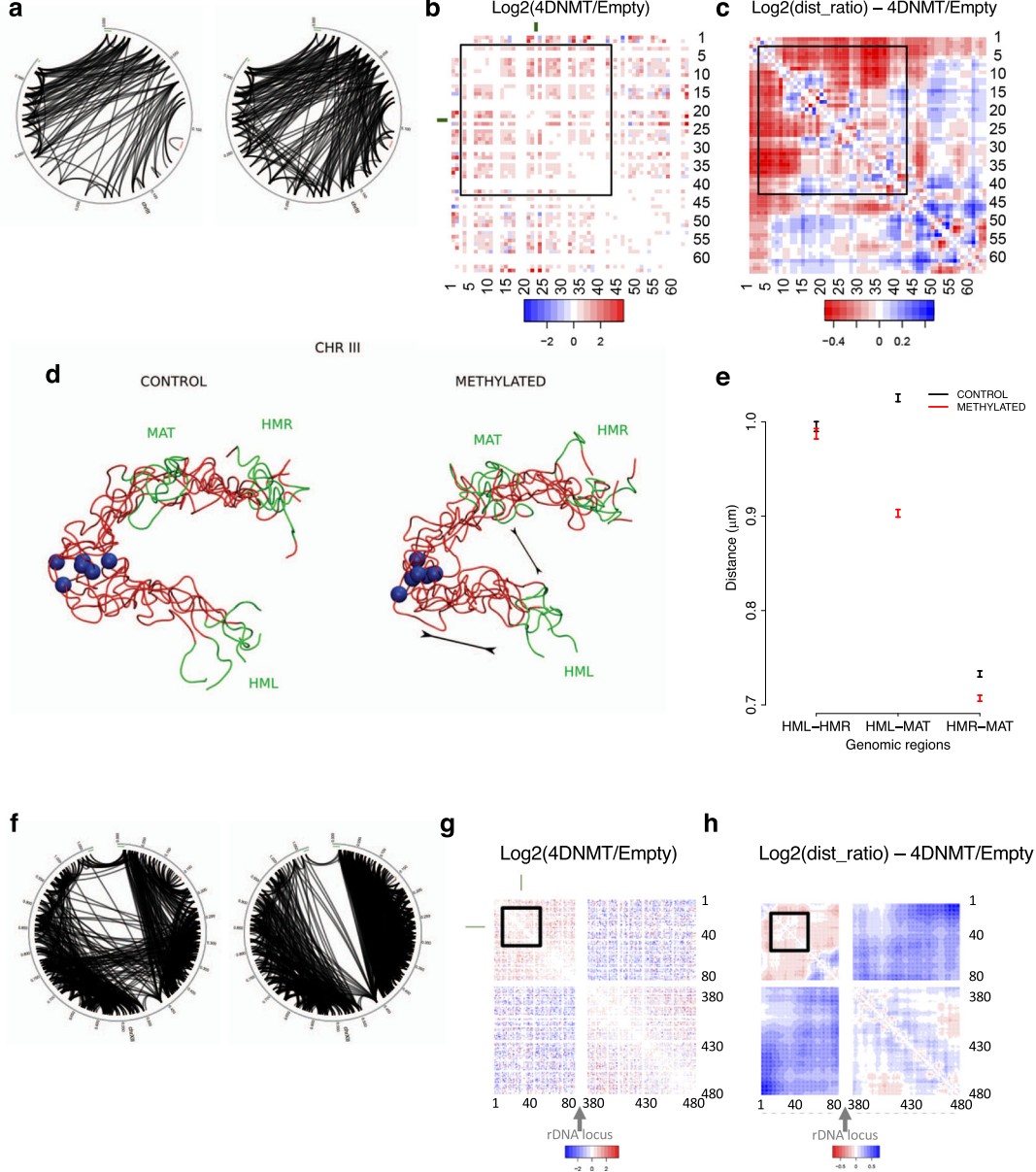

**Fig. 7 Chromosome conformation changes under DNA methylation. a** Circos diagrams of significant interactions in chromosome III for the control (left) and methylated (right) samples. Heatmaps displaying the log2 ratio (Methylated/Control) of (**b**) the contact frequencies and (**c**) the distances in the ensemble for chromosome III. Blue indicates interaction with a higher frequency or shorter distance in the non-methylated control sample and red indicates interactions with a higher frequency or shorter distance in the methylated samples. **d** 3D representative structures for chromosome III highlighting the distance between MAT and HML loci shown in green. The centromere is represented as a blue sphere. **e** Average distances (±standard error) between mating type loci in the ensemble of structures for chromosome III ($n = 260$ and 482 structures for control and methylated samples, respectively). **f** Circos diagrams of significant interactions in chromosome XII for the control (left) and methylated (right) samples. Log2 ratio (Methylated/Control) of (**g**) the contact frequencies and (**h**) the distances in the ensemble for chromosome XII.

for whole-genome bisulfite sequencing, RNA extraction for RNA-sequencing or Semi-intact cell preparation for MNase digestion and nucleosome mapping.

The yIL30 strain (Mata ade2-1 can1-100 his3-11, 15 leu2-3, 112 trp1-1 ura3-1 ade2-1::His3p-CFP-lacI-URA3p-λcIYFP-ADE2, TetR-mRFP:NAT1, MAT5':: λO-HIS, HMR:: LacO-TRP, HML::TetO-LEU) carrying the three integration sites for the fluorescent tags was kindly provided by Kerstin Bystricky.

The yIL30-W strain (Mata ade2-1 can1-100 his3-11, 15 leu2-3, 112 trp1-1 ura3-1 ade2-1::His3p-CFP-lacI-URA3p-λcIYFP-ADE2, TetR-mRFP:NAT1, MAT5':: λO-HIS, HMR:: LacO-TRP, HML::TetO-LEU, trp1Δ::kan) was obtained by replacing the endogenous *TRP1* locus in yIL30 by the kanamycin resistance gene amplified from pFA6A-kANr using the F2_pFA6aKan_TRP1 and R1_pFA6aKan_TRP1 oligonucleotides. For live cell imaging yIL30-W cells were transformed with the DNMT3a-DNMT3b-pBEVYGU and DNMT3L-DNMTT1-pBEVYGT plasmids or with the corresponding empty vectors. In both cases the

expression was controlled by the Gal1 promoter. For the analysis selected transformants were grown until saturation in synthetic medium with galactose and washed twice with SC-Low Fluorescence medium (lacking folic acid and riboflavine).

**Flow cytometry analysis.** 0.5 ml of culture ($OD_{600} = 0.6$–0.8) were collected and centrifuged for 5 min at $1000 \times g$ at RT. Pellets were washed twice with 1x ice-cold PBS and resuspended in 50 μl of 1x ice-cold PBS. 20 μl of cells were fixed with 1 ml of 70% EtOH overnight at 4 °C. Samples were washed with 1x Saline Sodium Citrate buffer (SSC; 150 mM NaCl, 15 mM Na citrate, pH 7.8 for 20x SSC). The pellet was resuspended in 0.5 ml of 1x SSC, treated with 0.5 mg/ml RNase A (Roche) for 1.5 h and then with 0.5 mg/ml Proteinase K (Roche) for another 1.5 h at 50 °C. After incubation, cells were briefly sonicated for 10 mn, medium potency,

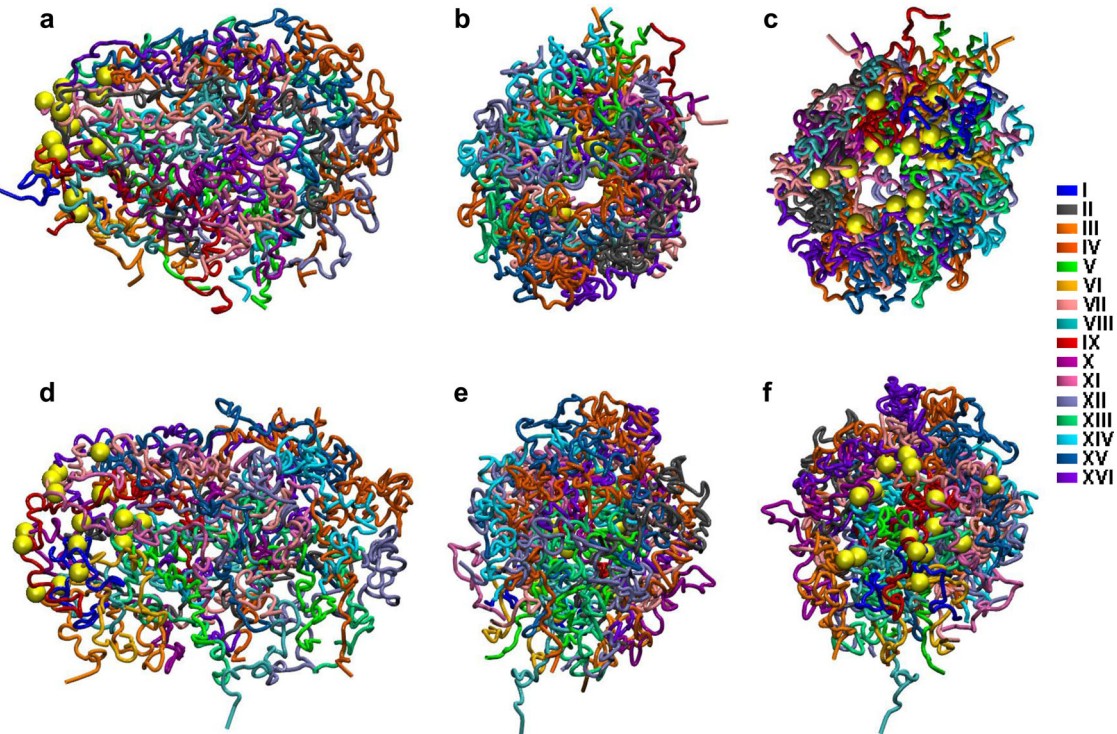

**Fig. 8 Whole-genome 3D model.** Three views are provided representing three different angles for (**a–c**) the control samples and (**d–f**) the methylated sample. Each chromosome is represented in a specific color. Centromeres are represented in yellow. The rDNA in chromosome XII is not represented due to the lack of information provided by the Hi-C in repeated regions.

by using the Bioruptor system (intervals of 10 s on-20 s off). 250 μl of the cells were added in 0.5 ml of 1x SSC containing 1 μM Sytox Green (Sigma) and were incubated 10–20 mn in the dark (room temperature) before analyzing the DNA content using a Beckam Coulter Gallios™ flow cytometer. MultiCycle software (Phoenix Flow Systems, San Diego, CA, USA) was used to determine the percentage of cells in the cell cycle phases.

**HPLC/MS.** HPLC/MS/MS analysis was based on the protocol described[58,54]. A Kinetez 2.6 μm HILIC 100 A column (150 mm × 4.6 mm) (Phenomenex) and a Acquity UPLC system (Waters Corp., Milford, MA, USA) coupled to a mass spectrometer API 3000™ (AB Sciex, Foster City, CA, USA) triple quadrupole working in MRM (multiple reaction monitoring) method in positive mode. Two eluents were used: eluent A2 (Acetonitrile) and eluent B1 (0.1 M ammonium formate adjusted at pH 3.2) with a isocratic gradient 8 min of total running time at 90% A and 10% B for the nucleosides elution. The separation was performed in a flow of 1400 μl min$^{-1}$, with 10 μl injection volume and two replicates each, totaling two biological replicates and two technical replicates of each sample. The standard nucleosides cytosine and methyl-cytosine (Sigma) were diluted in HCl 0.01 N and stored at −20 °C. The m/z transitions from 112 to 95 (cytosine) and from 126 to 81 (methyl cytosine) were chosen for MRM experiments. The peak area obtained was analyzed by Analyst 1.4.2 (AB Sciex). Quantification (%) was performed according to 5mdC concentration divided by 5mdC concentration plus dC concentration multiplied by 100.

**Western blot.** Proteins were extracted by resuspending the pellet of cells from a 20 ml cultures at OD600 = 1 in 400 μl of RIPA buffer (50 mM Tris pH7.5, 150 mM NaCl, 1% NP40, 0.5% NaDeoxycholate, 0.1% SDS) containing 1 mM PMSF and protease inhibitors (cOmplete ULTRA Tablets, Mini, EASYpack, Roche). 400 μl of glass beads were added and samples were processed using FastPrep (MP) for 3 times for 20 s pulses @4.5 m/s. After centrifugation 5 min at 2350 x g, supernatants were recovered and quantified by bradford. 20 μg of protein were loaded on a 6 or 8% acrylamide gel and subjected to PAGE, proteins were then transferred onto an immobilon membrane (millipore) for subsequent hybridization with anti-DNMT1 (ref ab87654, Abcam, dilution 1/1000), anti-DNMT3a (ab2850, Abcam, dilution 1/2000), anti-DNMT3b (ab122932, Abcam, dilution 1/500) or anti-Flag (F7425, Sigma, dilution 1/2000) antibody overnight followed by secondary antibody anti Rabbit (Goat)-HRP conjugated (65-6120, Invitrogen, dilution 1/5000). The signal was revealed using ECL™ prime WB detection reagent (Amersham, GE Healthcare).

**Ume6 purification.** Rosetta *E. coli* cells containing the *UME6*- pET14b plasmid were grown at 37 °C in 1 L of LB ampicillin medium supplemented with 1% of glucose up to OD$_{600nm}$ 0.6 expression was induced with 0.15 g/L IPTG at 20 °C for 4 h. Cells were harvested by centrifugation and pellet was resuspended in 20 ml of binding buffer (50 mM sodium phosphate ph7.4, 300 mM NaCl, 1 mM PMSF, 0.01% tween20, lysozyme 0.2 mg/ml, 1 mM MgCl2, 2000u DNase and a tablet of cOmplete™EDTA-free Protease Inhibitor Cocktail), incubated for 30 min on ice and sonicated twice for 4 min, followed by centrifugation at 10.000 × g for 15 min. The supernatant was incubated 10 min with the Dynabeads™ His-Tag Isolation and Pulldown (ThermoFisher). Beads were washed 4 times with wash buffer (50 mM sodium phosphate ph7.4, 300 mM NaCl, 5 mM imidazole, 0.01% tween20). Finally, protein was eluted with the elution buffer (50 mM sodium phosphate ph7.4, 300 mM NaCl, 300 mM imidazole, 0.01% tween20) and dialyzed overnight with 10 mM sodium phosphate pH 7.4, 50 mM NaCl and 10 μM ZnCl2 buffer. Ume6 protein was visualized by SDS-PAGE and coomassie brilliant blue staining.

**Gel retardation assay.** The double stranded (ds) probes containing the URS1 of *SPO13* with or without methylated cytosines was obtained by annealing the following forward and reverse oligonucleotides: Spo13-Cy5-F with Spo13-R, and Spo13-Met-Cy5-F with Spo13-Met-R, respectively (Supplementary Table 4).

The reactions were incubated for 5 min at 95 °C and cooled down overnight. The resulting ds DNA fragments were then purified from 2% agarose gel.

Ume6 protein in 10 mM NaPO$_4$ pH 7.5, 50 mM NaCl and 10 μM ZnCl$_2$ was diluted with 1 μg/μl BSA. Salt concentration for each reaction was adjusted to a final concentration of 115 mM NaCl. For the binding reactions the protein samples were maintained in the same buffer and 168 nM of fluorescent Cy5 URS1 of *SPO13* gene with or without methylated cytosines was added. Reactions were incubated for 15 min at 20 °C and 20% of glycerol was added before loading into 8% polyacrylamide gel in Tris-Borate-EDTA buffer. Electrophoresis was performed for 1 h at 120 V. Band intensities were obtained by exposing the gel to Typhoon phosphorImager scanner equipped with the emission filter for Cy5 fluorescence. The intensity of the bands was quantified using the software ImageQuant TL v.8.2.0.

**DNA methylation pattern.** Whole-genome bisulfite sequencing (WGBS) – WGBS was performed following the procedure outlined in[9]. Briefly, genomic DNA (1–2 μg) was spiked with unmethylated λ DNA (5 ng of λ DNA per μg of genomic DNA) (Promega). The DNA was sheared by sonication to 50–500 bp using a Covaris E220 and fragments of size 150–300 bp were selected using AMPure XP beads (Agencourt Bioscience Corp.). Genomic DNA libraries were constructed using the Illumina TruSeq Sample Preparation kit (Illumina Inc.) following the lllumina standard protocol: end repair was performed on the DNA fragments, an

adenine was added to the 3' extremities of the fragments and Illumina TruSeq adapters were ligated at each extremity. After adapter ligation, the DNA was treated with sodium bisulfite using the EpiTexy Bisulfite kit (Qiagen) following the manufacturer's instructions for formalin-fixed and paraffin-embedded (FFPE) tissue samples. Two rounds of bisulfite conversion were performed to assure a conversion rate of over 99%. Enrichment for adapter-ligated DNA was carried out through 7 PCR cycles using the PfuTurboCx Hotstart DNA polymerase (Stratagene). Library quality was monitored using the Agilent 2100 BioAnalyzer (Agilent), and the concentration of viable sequencing fragments (molecules carrying adapters at both extremities) estimated using quantitative PCR with the library quantification kit from KAPA Biosystem. Paired-end DNA sequencing (2x100bp) was then performed using the Illumina Hi-Seq 2000.

Read mapping and estimation of cytosine methylation levels – The WGBS reads were processed using the gemBS pipeline v3.0[59] using as reference S. cerevisiae S288c. Reads with MAPQ scores < 20 and read pairs mapping to the same start and end points on the genome were filtered out after the alignment step. The first 5 bases from each read were trimmed before the variant and methylation calling step to avoid artifacts due to end repair. For each sample, CpG sites were selected where both bases were called with a Phred score of at least 20, corresponding to an estimated genotype error level of ≤1%. Sites with >500x coverage depth were excluded to avoid centromeric/telomeric repetitive regions. CpGs were considered methylated when the number of mapped reads was larger than 10 and the estimated methylation percentage was above 0.1.

DNMT specificity analysis – We extracted two bases downstream and upstream from each CpG (having at least ten WGBS reads mapped) and trained a logistic regression model (using R) for the number of converted and non-converted Cs, using the extracted motifs as predictors for each WGBS sample (samples removing one of the DNMTs, T859, T860, T861 and T869; and two samples with the four DNMTs, T862 and T863). We computed for each sample the effect of each motif and its standard deviation, and used it to determine those with a significant effect on methylation level (estimated effect above two standard deviations). We found motifs specific for each sample lacking one of the DNMTs (motifs with significant effect in the sample removing one DNMT but not significant in the sample with all DNMTs) and compared their relative frequencies in all samples.

Nanopore sequencing – Suspensions of spheroplasts from methylated and control S. cerevisiae strains were loaded on Sage Science gel cassettes to perform lysis under electrophoretic conditions. DNA content in each sample was estimated by the cell count. A number of spheroplasts equivalent to 10 µg of genomic DNA were resuspended in 70 µl of HLS Suspension buffer (Sage Science, Mammalian white Blood cell suspension kit, #CEL-MWB1) and loaded on the gel cassettes (Sage Science, SageHLS HMW DNA extraction kit #HEX-0012).

The custom Sage HLS (Sage Science) protocol used (Extraction Collection DC55V 1h15m) was accommodated for the yeast small chromosome sizes. This custom protocol did not include a DNA fragmentation step. In brief, during the extraction step, the High Molecular Weight (HMW) yeast gDNA was bound in agarose while the solubilised and degraded proteins and other contaminants were kept in solution. The Sage Science Buffer A was used as a lysis buffer for this step. In the last step of the protocol, the HMW DNA was retrieved from the gel through an automated elution process that was optimized to elute all the yeast chromosomes in the elution module number 2 of the cassette.

Elution modules 1, 2, 3 & 4 were selected for the library preparation of the control and methylated S. cerevisiae samples. For each condition, the selected elution modules were pooled, purified with 1-fold excess of Agencourt AMPure XP beads (Beckman Coulter, A63882) and eluted in water. Two barcoded libraries containing both type of samples were prepared using the Oxford Nanopore Ligation sequencing kit (ONT, SQK-LSK109) combined with the Oxford Nanopore Native Barcoding Expansion kit (EXP-NBD103 1D) following manufacturer's instructions.

After connecting the flows cells to the MinION Mk1b device, the MinKNOW interface QC (Oxford Nanopore Technologies) was run in order to assess the flow cell quality. Once the priming of the flow cell was finished, from 200 ng to 600 ng of the final barcoded library was loaded into R9.4.1 FLO-MIN106 or FLO-MIN106D flow cells and the sequencing data were collected during 48 h. The quality parameters of the sequencing runs were further monitored by the MinKNOW platform in real time. The MinKNOW versions used was 1.15.4. The basecalling was performed using Guppy 2.3.7. Reads were mapped using minimap2 2.9-r720, and CpG methylation was called using nanopolish 0.11.0.

**Nucleosome mapping.** Semi-intact Yeast cell preparation – Semi-intact cells were prepared as previously described[60]. Briefly, cells were grown at 30 °C in 300 ml YPD to =1 ×107 cells/ml. For each 250 ml of cells (10⁷cells/ml), semi- intact cells were prepared as follows. Cells were collected by centrifugation (700 × g, 7 min, RT), resuspended in 25 ml 100 mM Pipes, pH 9.4, 10 mM DTT, incubated with gentle agitation at 30 °C for 10 min, and collected by centrifugation (1000 × g, 5 min, RT). Cells were resuspended in 6 ml YP, 0.2% glucose, 50 mM KPO4, pH 7.5, 0.6 M sorbitol. 10 u zymolase was added, and the suspension was incubated with gentle shaking 30 °C for 30 min. Spheroplasting was monitored by light microscopy. Great care was taken not to overdigest cells to avoid lysis. Spheroplasts were collected by centrifugation at 1000 × g for 5 min at RT, re- suspended with a plastic pipette in 40 ml YE 1% glucose, 0.7 M sorbitol, and incubated with gentle shaking at 30 °C for 20 min. Spheroplasts were collected by centrifugation (1000 ×

g, 5 min, RT) and washed twice at 4 °C with cold permeabilization buffer (20 mM Pipes-KOH, pH 6.8, 150 mM K-Acetate, 2 mM Mg-Acetate, 0.4 M sorbitol. The final pellet was resuspended in 1 ml cold permeabilization buffer containing 10% (v/v) DMSO. 100 µl aliquots were placed in 1.5 ml microfuge tubes and frozen slowly above liquid N₂ and stored at −80 °C.

MNase-seq – 0.4 ×109 semi-intact cells were digested with micrococcal Nuclease (MNase), 1.5 unit at 37 °C for 30 min with 3 mM CaCl2. The reactions were stopped by addition of EDTA to a final concentration of 0.02 M and subsequently incubated with RNase A (0.1 mg) for 4 h at 37 °C and further treated with Proteinase K at 37 °C o/n. DNA was purified using phenol–chloroform extraction and concentrated by ethanol precipitation.

The percentage of mononucleosomal DNA fragments was examined by 2% agarose gels. Furthermore, the integrity and size distribution of digested fragments were determined using the microfluidics-based platform Bioanalyzer (Agilent) prior to library preparations following Illumina standard protocol. The short-insert paired-end for MNase sequencing were prepared with PCR free protocol using KAPA Library Preparation kit (Roche). In short, 2.0 micrograms of Micrococcal nuclease (MNase) digested genomic DNA from S. cerevisiae was end-repaired, adenylated and ligated to Illumina platform compatible adapters with dual indexes (Integrated DNA Technologies). The adapter-modified end library was size selected and purified with AMPure XP beads (Agencourt, Beckman Coulter). The final libraries were quantified by Kapa Library Quantification Kit for Illumina platforms (Roche).

The libraries were sequenced using TruSeq SBS Kit v4-HS (Illumina), in paired-end mode with a read length of 2x76bp following the manufacturer's protocol. Images analysis, base calling and quality scoring of the run were processed using the manufacturer's software Real Time Analysis (1.18.66.3).

Nucleosome calling – MNase-seq paired-end reads were mapped to yeast genome (sacCer3, Apr. 2011) using Bowtie[61] aligner, allowing a maximum of 2 mismatches and maximum insert size of 500 bp. Output BAM files were imported in R[62,58] and quality control was performed with htSeqTools package to remove PCR artifacts[63]. Filtered reads were processed with nucleR package[64] as follows: mapped fragments were trimmed to 50 bp maintaining the original center and transformed to reads per million. Then, noise was filtered through Fast Fourier Transform, keeping 2% of the principal components, and peak calling was performed using the parameters: peak width 147 bp, peak detection threshold 35%, maximum overlap of 80 bp, dyad length 50 bp. Nucleosome calls were considered well-positioned when nucleR peak width score and height score were higher than 0.6 and 0.4, respectively, and fuzzy otherwise.

Nucleosome Dynamics – NucDyn R package[45] was used to find changes in nucleosome organization between control and methylation-induced samples. P-values quantifying the nucleosome change were obtained running NucDyn with the following parameters: maximum difference of 70, maximum length of 140, minimum number of reads to report a shift of 3, shifts threshold of 0.1, indels minimum number of reads to report evictions and inclusions (indels) of 3, indels threshold of 0.05.

Genomic Annotation – Data was annotated from the UCSC gene track that contains 6692 genes. We discarded genes that are described as "Putative" or "Dubious" and genes located in the mitochondrial chromosome. We used gene lengths to normalize methylation proportions, nucleosome coverages and CpG density partitioning each gene in 137 bins (each bin has on average 10 bp since the mean length of yeast genes is 1369 bp).

**Gene expression.** mRNA library preparation and sequencing – The RNASeq libraries were prepared from total RNA (extracted by the standard hot phenol protocol) as follows. Total RNA quality and quantity were assessed using Qubit® RNA HS Assay (Life Technologies) and RNA 6000 Nano Assay on a Bioanalyzer 2100 (Agilent). The RNASeq libraries were prepared following KAPA Stranded mRNA-Seq Illumina® Platforms Kit (Roche) following the manufacturer´s recommendations. Total RNA (500 ng) was enriched for the polyA mRNA fraction and fragmented by divalent metal cations at high temperature. In order to achieve the directionality, the second strand cDNA synthesis was performed in the presence of dUTP. The blunt-ended double stranded cDNA was 3´adenylated and Illumina platform compatible adapters with unique dual indexes and unique molecular identifiers (Integrated DNA Technologies) were ligated. The ligation product was amplified with 15 PCR cycles and the final library was validated on an Agilent 2100 Bioanalyzer with the DNA 7500 assay (Agilent).

The libraries were sequenced on HiSeq2500 (Illumina) using TruSeq SBS Kit v4-HS (Illumina), in paired-end mode with a read length of 2x76bp following the manufacturer's protocol. Images analysis, base calling and quality scoring of the run were processed using the manufacturer's software Real Time Analysis (1.18.66.3). We generated over 20 million paired-end reads for each sample in a fraction of a sequencing lane.

RNA-seq data processing and analysis – RNA-seq reads were mapped against the yeast reference genome (Sacharomyces_cerevisiae.R64-1-1+plasmid) using STAR version 2.5.2a[65] with ENCODE parameters. Quantification of annotated genes (ensembl release 87) was done using RSEM version 1.2.28[66] with default options. Heatmaps with the top differentially expressed genes was perform with the pheatmap R package. Differential expression between conditions was performed with DESeq2 version 1.18[67] with default parameters.

Motif enrichment analysis – De novo promoter motif enrichment for genes with highly methylated promoters and increase in expression level was performed using HOMER[68]. It uses a cumulative hypergeometric distribution to score motif enrichment in the target set compared to the background set. Target set was defined in the region 200 bp downstream the TSS.

**ChIP-seq.** Sample preparation – Yeast strains were grown into the appropriate selective media with 2% galactose and 1% raffinose to stationary phase ($OD_{600}$ similar to 8), diluted to $OD_{600} = 1$ in 50 ml of culture media and crosslinked 20 min with 1% of formaldehyde followed by a 15 min incubation with 125 mM of glycine. After crosslinking, spheroplasts were isolated as described before using zymolase and resuspended in 0.3 ml of lysis buffer (50 mM Hepes-KOH at pH 7.2, 140 mM NaCl, 1 mM EDTA, 0.1% Deoxycholic acid sodium salt and 1% Triton X-100) containing a cocktail of protease inhibitors (EDTA-free Tablets, Roche). An equal volume of glass beads (0.5-mm diameter) was added, and the spheroplasts were broken using a bead-beater (FastPrep-24, Biomedicals). Glass beads were then removed and the lysate was transferred to a Sorenson tubes to digest the chromatin into fragments of 300 nucleotides using the Bioruptor Pico (30 cycles, 30"on/30"off). The whole extract was clarified by centrifugation for 10 min at $5000 \times g$ at 4 °C and an aliquot was taken as input. In parallel, 50 μl of Dynabeads M-280 Sheep anti Rabbit IgG (Thermo Fisher) per sample were washed twice with PBS + 5 mg/ml of BSA and incubated overnight at 4 °C with 2.5 μg of the primary antibody (H3K4me1 (ab8895, Abcam), H3K4me3 (ab8580, Abcam)). Beads were then washed with PBS + 5 mg/ml of BSA and resuspended in 30 μl/sample of PBS-BSA 5 mg/ml. Extracts were then incubated 2 h at 4 °C with the Dynabeads, previously conjugated with the primary antibody, and then washed two times with lysis buffer, two times the lysis buffer supplemented with 360 mM NaCl, 2 times with washing buffer (0.5% Deoxycholic acid sodium salt, 10 mM TRIS pH8, 250 mM LiCl, 0.5% NP-40, 1 mM EDTA) and one time with TE (10Mm Tris-HCl pH 7.5,1 mM EDTA). Dynabeads were resuspended in 80 μl of Elution Buffer (50 mM TRIS, 10 mM of EDTA, 1% SDS) and crosslinking was reverted incubating overnight at 65 °C followed by 2 h incubation at 37 °C with 0.80 mg/ml per sample of proteinase K. DNA was purified by phenol–chloroform extraction and ethanol precipitation. The libraries from ChIPseq were done using the NEBNext Ultra DNA Library Prep kit for Illumina (Ref.:#7370) following the manufacturer protocol and sequenced at 1x50bp.

ChIP-seq analysis – Sequenced reads were analyzed using Galaxy platform[69]. Reads were mapped to the reference sacCer3 genome with BWA[70] and fragments with MAPQ quality score below 20 were discarded. Peak calling was performed using MACS2[71] with default parameters.

**3D genome structure.** Hi-C libraries – The protocol was performed as previously described[72] with a few modifications. 100 ml of yeast culture were crosslinked with 3% formaldehyde during 20 min and quenched with Glycine 125 mM during 5 min at RT. Cells were crushed during 30 min in liquid nitrogen and the chromatin was digested with HindIII. The DNA overhangs were filled-in with dNTP including Biotin-14-dATP, and the resulting blunt end were ligated. After ligation, samples were purified with phenol:chloroform and DNA was precipitated with ethanol.

The paired-end Hi-C-sequencing libraries were prepared with KAPA Library Preparation kit (Roche) with some modifications. The biotin marked and de-crosslinked DNA was sheared to a size of 300–500 bp on Covaris™ LE220 (Covaris) focused-ultrasonicator. The fragmented DNA was end-repaired, adenylated and the biotin-tagged DNA was pulled down using the Dynabeads™ MyOne™ Streptavidin C1 beads (Thermo Fisher Scientific). The biotinylated fragments were ligated to Illumina platform compatible adapters with unique dual indexes and unique molecular identifiers (Integrated DNA Technologies) and enriched by 12 PCR cycles by KAPA HiFi PCR Kit (Roche).

The libraries were sequenced on HiSeq2500 (Illumina) using TruSeq SBS Kit v4-HS (Illumina), in paired-end mode with a read length of 2x76bp following the manufacturer's protocol. Images analysis, base calling and quality scoring of the run were processed using the manufacturer's software Real Time Analysis (1.18.66.3).

Hi-C data processing and normalization – We processed Hi-C data using TADbit[73] [https://github.com/3DGenomes/tadbit] for quality control, mapping and filtering. First, quality control was performed with the FastQC protocol implementation in TADbit. Then, reads were mapped to the reference yeast genome (sacCer3, Apr. 2011) with a fragment-based strategy. Afterwards, non-informative contacts (self-circle, dangling-end, error, duplicated and random breaks) identified by TADbit were filtered-out, obtaining 32–37 million valid interactions per experiment. Off-target contacts (neither end of the read mapped to one of the capture regions) were also discarded (full details of the number of excluded reads are given in Supplementary Table 5). Finally, contact matrices were created from valid reads at 5 kb resolution with the corresponding TADbit module, and low frequency bins were removed.

Contact matrices were transformed to .hic format for visualization in Juicebox[74] using the pre command, and normalized with the Balanced method[75].

Differential Hi-C analysis was performed using the R/Bioconductor package diffHic[76]. The mapped Hi-C data were filtered and the differential interaction analysis between the control and methylated samples (using the two replicates for each treatment) was performed using the procedure recommended in the diffHic manual.

Hi-C-based chromatin 3D structure – High resolution Hi-C data at 5 kb was used to obtain the 3D structure, conformation and dynamics of entire yeast chromosomes and their context inside the nucleus. The Hi-C technique provides interaction contacts between DNA fragments. The interaction counts or frequencies between two loci $i$ and $j$ ($f_{ij}$) can be converted to spatial 3D distances between those loci ($d_{ij}$) by an inverse relationship (Eq. (1)),

$$d_{ij} = \gamma / f_{ij}^{\alpha} \tag{1}$$

where $\gamma$ represents the scale of the structure and is usually taken to match experimental distances between selected genomic regions, and the precise value of $\alpha$ depends on the organism under study, the genomic distance, and the resolution of the Hi-C map and needs to be fitted[77–79]. In the present work, $\gamma$ was taken for the model to match the size of the cell nucleus measured by confocal microscopy and $\alpha$ was fitted to maximize the correlation between experimental and modeled contact maps.

Since Hi-C interaction counts are known to present several biases, such as mappability of fragments, GC content, and fragment length, they were normalized using iterative correction and eigenvector decomposition[80]. Finally, the output of the conversion procedure was a matrix containing equilibrium distances ($r_0$) for the different interacting loci. To remove the background noise, a cutoff of two times the median of all trans contacts (i.e., between different chromosomes) was applied to the HiC contact map to define interacting regions.

The chromosome model was built as a chain of beads, each bead representing a genomic region that corresponds to a bin from the Hi-C map. Spatial equilibrium distances were obtained from Eq. (1) as explained above. The distances between interacting beads ($r$) were restrained near their equilibrium length during the simulations by penalizing with a harmonic potential (Eq. (2)) when approaching at shorter distances or moving away at longer distances than the equilibrium. A tolerance of one bead radius was applied, thus resulting in a flat-welled parabola potential (Eq. (2)),

$$E = k(r - r')^2 \tag{2}$$

where $r' = r_0 - r_{bead}$ for $r < r_0 - r_{bead}$, $r' = r$ when $r$ lies within $r_0 \pm r_{bead}$ and $r' = r_0 + r_{bead}$ when $r > r_0 + r_{bead}$.

To ensure proper connectivity of the fiber, consecutive beads were also bound by a harmonic potential but with a force constant five orders of magnitude stronger than that applied to interacting non-consecutive beads. An excluded volume was defined for each bead by a standard Lennard-Jones potential with equilibrium distance equal to one bead radius and a soft energy well. Additional repulsive restraints were added for non interacting beads, forced to remain at a distance longer than the maximum equilibrium distance obtained from Eq. (1). The initial structure of the chromosome fiber was varied between an extended conformation and a random localization of initially unbound beads in different replicas. The system was allowed to sample the conformational space using pmemd simulation engine for GPU from Amber 18 package. Different conformations of the fibers were determined by attraction and repulsion forces arising from the distance restraints between beads.

In the end, an ensemble of structures was obtained by selecting the minimum number of snapshots minimizing the number of experimental restraint violations (equilibrium distances input). This method yields a population of structures with different conformations, which in average, but not individually, reproduce experimental Hi-C maps derived from population of cells with variable chromatin structure[81].

The ensemble was built in the following way. First, sampled structures with more restraint violations than the mean restraint violations were discarded. Then, the structure with less restraint violations was selected. Considering only the restraints violated by the selected structure, the structure fulfilling more of these restraints was kept. The procedure was repeated iteratively, always considering the restraints that were not fulfilled by any of the previously selected structures. Iterations were stopped when there was no structure left fulfilling new restraints.

Chromatin coarse-grained model at the nucleosome level – The starting point for the 3D chromatin model at the nucleosome level is the coverage of the MNase-seq signal obtained using NucleR software[64]. Different families bearing nucleosomes in locations compatible with the MNase-seq experiment are derived by deconvolution of the coverage signal by using a composite Gaussian approximation. For each of the resulting families (compatible with the MNase-seq signal and DNA/histone stoichiometry) an ideal 3D chromatin structure is prepared and further simulated by a coarse-grained Monte Carlo sampling approach with flexible linkers and rigid nucleosomes. Linker DNA is represented at the base pair level by a pseudo-harmonic potential expressed in helical parameters (rise, slide, shift, twist, roll, tilt)[82]. Debye Huckel electrostatics and excluded volume potentials were added to avoid overlaps (exact details of the simulation procedure will be described elsewhere). The results of the different simulations are clustered to select the minimum number of nucleosome structural families that makes physical sense and that together reproduce Mnase-seq experiments. The source code and instructions to run simulations are available as Supplementary Software 2.

**Microscopy and image analysis.** yIL30-W cells expressing the 4DNMTs were attached to 35 mm glass bottom culture dishes coated with Concanavalin A[83]. High-content image acquisition was performed with an Automated Wide-field Olympus IX81 Microscope, an MT20 xenon mercury lamp (Olympus Life Science Europe,

Waltham, MA) and a 100x/1.3 oil UPlanFL N objective. CFP, mRFP and YFP were acquired using the appropriate filter sets for fluorescence imaging (AHF Analysentechnik, Tuebingen, Germany). ScanR Acquisition software version 2.3 was used to automatically record several fields of view and to take z-stacks (21 slices with a z-step of 200 nm) at every position. At least 200 cells from each condition were analyzed. For the image analysis, the "Spot distance" [http://bigwww.epfl.ch/sage/soft/spotdistance/] plug in Version 08.2014 (Daniel Sage, EPFL) on ImageJ Fiji distribution[84] was used to find the coordinates of every focus and to get the distances among them. Statistical analysis was performed using the Wilcoxon test.

**Reporting summary**. Further information on research design is available in the Nature Research Reporting Summary linked to this article.

## Data availability

The data that support this study are available from the corresponding authors upon reasonable request. All sequencing data generated in the course of this study have been deposited in the European Nucleotide Archive (ENA). Raw data have been deposited in the European Nucleotide Archive (ENA) under the following accession numbers and hyperlinks: WGBS: E-MTAB-9258. RNA-seq: E-MTAB-9195. MNase-seq: E-MTAB-9259 and E-MTAB-10254. Hi-C: E-MTAB-9257. Nanopore processed data as well as FASTQ data have been deposited in the ENA with accession number E-MTAB-9356. ChIP-seq raw and processed data have been submitted to the European Nucleotide Archive (ENA) under accession number E-MTA-10001. Source data for Western blots (Supplementary Fig. 1) and for gel and image quantifications (Supplementary Figs. 10 and 18) are provided with this paper. Source data are provided with this paper.

## Code availability

Programs for reconstruction and visualization of chromatin structure (from nucleosome to entire chromatin) are available at the H2020 MuG [https://www.multiscalegenomics.eu/] virtual research environment. The code used to identify the differentially methylated populations of nanopore reads is provided as a compressed file (Supplementary_Software1-zip), with the program, 3 test data files and a short README, available as Supplementary Software 1. The code to run simulations for the Coarse Grained model is provided as a compressed file (Supplementary_Software2.zip) available as Supplementary Software 2.

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

## Acknowledgements

We thank all the former and actual members of the EBL for technical support and helpful discussions. We acknowledge the IRB Biostatistics Core Facility for their help with the ChIP-seq analysis and Anna Lladó Equisoain from the IRB Advanced Digital Microscopy Facility for her help with live-cell imaging. We also want to thank Ron Schuyler, Mike Goodstadt, François Serra and David Castillo (CRG-CNAG), F.Posas, E.Nadal and F. Azorin (IRB) and I. Pazos and O. Gallego (UPF) for fruitful discussions and technical advices. We are grateful to Dr Jessie Colin for providing various expression vectors, to Dr Jan Fronck, Dr Shen Li and Dr Jia-Lei Hu for providing *DNMT1, DNMT3a, DNMT3b* and *DNMT3L* cDNA and to Kerstin Bystricky for sharing the yIL30 strain and the method to monitor chrIII architecture. This work has been supported by the Spanish Ministry of Science (BIO2012-32868), the Catalan SGR, the Instituto Nacional de Bioinformática, the European Research Council (ERC_SimDNA) and the BioExcel and MuG VRE H2000 projects. M.O. is an ICREA Academia Fellow. This project also received funding from the European Union's Horizon 2020 research and innovation programme under the Marie Skłodowska-Curie grant agreement No. 754510 [to J.P.A. and M.O.]. The work of S.H. was supported by the Spanish Ministry of Science (PGC2018-099640-B-I00). A.E.C. is funded by ISCIII /MINECO (PT17/0009/0019) and co-funded by FEDER.

## Author contributions

M.O. and I.B.H. designed the study; M.L., R.L., D.Be., I.B.H. and N.V. performed in vivo and in vitro experiments; D.Be. performed HPLC/MS analysis; D.Bu., O.F., A.E.-C., J.P.A. and S.C.H. performed data analysis; D.Bu., J.P.A. and P.D.D. performed in silico molecular simulations; J.B., M.G. and I.G.G. realized the sequencing; I.B.H., S.C.H., J.P.A. and M.O. wrote the manuscript.

## Competing interests

The authors declare no competing interests.
