## [Peer Review File · Nature Communications]

REVIEWER COMMENTS

Reviewer #1 (Remarks to the Author):

Comments for 262488_0_merged_1592425197

The manuscript entitled “Impact of DNA methylation on 3D genome structure” by Buitrago D., Labrador M., et al. addresses the relationship between the DNA methylation mark per se and chromatin function/architecture. The authors were able to reconstruct the mammalian methylation machinery expressing four murine proteins in yeast: de novo DNA Methyltransferases (DNMT3a and DNMT3b), maintenance DNA Methyltransferase (DNMT1) and a non-catalytic accessory factor (DNMT3L). The authors were able to achieve higher levels of 5meC than previous studies and they were able to recapitulate the results found in literature, confirming the validity of the model. Their main findings are:

- Introduction of DNA methylation in yeast generates a similar pattern as mammalian cells upon expression of two de novo DNA Methyltransferases, a maintenance methyltransferase and an accessory factor. They were able to determine a DNMT3-specific intrinsic sequence specificity.
- They recapitulated previous findings that DNA Methylation is mainly found between nucleosomes and it can influence nucleosome positioning, transcription factor binding and gene expression.
- DNA Methylation induces an increase in intra-chromosome contacts while decreasing inter-chromosome contacts, suggesting that RNA pol III transcribed loci can be involved in the organization.

Comments:

- 1) My main concern about the manuscript is that the authors do not consider that the presence of the DNA Methyltransferases themselves (2 de novo, 1 accessory factor, and 1 maintenance) could cause nucleosome fuzziness and/or rearrangements of the 3D genome structure, which are two of their main findings. It is well-established in the literature that DNMT3a/b/L interact with histone tails and they might have multimerization potential. The maintenance DNMT1 can also be considered as a reader of hemimethylated CpG sites. A possible solution would be to express in yeast:
 - A catalytic dead version for both active de novo DNMT3s (point mutant), in order to rule out the above mentioned DNMT3-dependent effect;
 - the bacterial CpG methyltransferase M.SssI, if they want to understand the effects of 5meC per se. The MTase should not be affected by histone modifications, but it should be blocked by the presence of nucleosomes;
- 2) A second point is that, based on the emphasis given in the title, I expected more data on genome structure. I suggest performing experiments with an orthogonal technique to validate the HiC results on a few loci that show differences between the control and the DNA methylation strain. In an ideal world, it would be through FISH/CRISPR-imaging, SPRITE or GAM.
- 3) Previous manuscripts were able to predict the abundance of DNA methylation or DNMT3 occupancy

based on the presence of a few chromatin marks genome-wide. Can the presence of a DNMT3-specific target DNA sequence improve the prediction?

4) Minor comments:

- Add page numbers

- Abstract:

- DNA methyltransferases, not DNA Methyl Transferases;
- *S. cerevisiae* should be in italics

- Introduction:

- "...that DNA methylation makes DNA less flexible and..." should be "...that DNA methylation makes the DNA less flexible and...";

- in the sentence "*S. cerevisiae* does not have methylation/demethylation machinery and no methylated DNA binding domain has been characterized", please specify that yeast lacks the DNA methylation machinery, since it possesses other methyltransferases;

- The Addition of DNA methylation machinery..., instead of "Addition of methylation machinery...";

- Results:

- Supplementary Fig. S2A: how many replicates for each time point/conditions? Please add the information

- It seems that I cannot find the DNA Methylation levels measured by HPLC/MS

- Supplementary Figures S3C/S4: I suggest to differentiate the Illumina-WGBS and Nanopore-seq tracks with different colors

- For the sentences: "For the cells synchronized in G1 we can see that ~50% of CpG sites have <5% methylation, but there are almost 20% of sites with methylation over 20%, and ~5% of sites with methylation over 50%. For the cells at saturation the methylation levels are generally higher, with 50% of sites having >15% methylation and 25% of sites having methylation over 50%.", I believe it will be easier for the reader to show an additional plot with the number of sites (or percentages) and the categories of DNA methylation levels you mentioned (<5%; 5-20%; 20-50%; >50%).

- Sentence: "The presence of two distinct populations of reads in the samples in exponential phase suggests that the DNA methylation maintenance machinery maybe not fully functional, leading to a difference in average methylation between the original and daughter DNA strands". Another hypothesis is that the DNA Methylation machinery is indeed fully functional, but it's not 100% effective due to the rapid replication of the genomic DNA in yeast during the exponential phase. Please articulate.

- Figure 1A,B: Please indicate that the plots are showing DNA Methylation levels from "replicating cells, synchronized in G1". Please add the y-axis (methylation levels/fraction).

- Figure 1G: Please differentiate with colors the stationary vs. G1 tracks.

- For the DNA methylation and Nucleosome positioning section, it would be great to cite NOME-seq, an experimental procedure to map DNA Methylation and nucleosomes on the same molecule.

- Figure 5A: maybe specify in the caption that the methylation probability is the shaded area. I would also set 0.4 as the maximum for the 5meC y-axis to zoom into the signal a little bit

- Figure 5B/C and section discussing DNA Methylation and transcription. The authors state "A differential expression analysis (Fig. 5B) shows that genes which are very lowly methylated do not

change their expression level between the control and transformed cells, while high methylation levels lead to important changes in gene activity in both directions: towards greater and lower expression (Fig. 5C and Supplementary table S1).” The changes in gene expression could also be due to the presence of the DNA methyltransferases themselves on the chromatin, not only to the DNA methylation mark alone. Please take this into account and modify the sentence accordingly. You can just state that there is a trend/correlation.

- Figure S9: You should also plot, as a control, the methylation and expression fold changes for a set of genes regulated by a specific transcription factor (non-overlapping with Ume6p) and one for randomly picked genes (a subset or all).

- Discussion:

- The authors state “Again, these differences are intrinsic and not coupled to any specific directing mechanism, which demonstrate the intrinsic ability of methylation to alter cellular phenotype.”: as mentioned by the authors and other reports, the underlying chromatin structure (presence of nucleosomes and histone modifications) dictates the possible substrates for the DNA Methyltransferases. It is correct that there are no factors that are specifically directing DNMT3s to a specific sequence. Please clarify the meaning of “which demonstrate the intrinsic ability of methylation to alter cellular phenotype”.

Reviewer #2 (Remarks to the Author):

Buitrago et al. report an interesting study on the effect of expression of the mouse DNA methylation machinery in yeast. The goal of this approach is to study the “intrinsic” preferences of the DNA methylation machinery (and their effects) on genome structure and function. In this sense, the overall approach seems most reminiscent of past work exploring in vitro nucleosome position and DNA methylation patterns, but this has the added advantage that this study is carried out in vivo and this can be used to study the relationship with other biological processes in the nucleus. The authors find that the introduction of the 4 DNMT proteins from mice leads to increased levels of DNA methylation (dependent upon conditions), changes in nucleosome patterning, 3D genome structure and some gene expression alterations. Further they characterize the patterns of DNA methylation deposition and correlate these with features the genome, such as the status of genes. Overall, I think that study will invite a criticism of lacking physiological relevance, but I think this misses the point, as I think the most clear comparison for this study have been past in vitro work. I find the manuscript interesting and well written. However, there are still several concerns that I have that I hope the authors can address before I think it would be appropriate for publication in Nature Communications.

Major points:

Perhaps the single biggest concern that I have is that while I find the data compelling, I am concerned that the patterns that they observe could be the result of growth/selection instead of innate preferences of the DNA methyltransferase activity. In other words, perhaps the observation of a low

level of DNA methylation near the promoters of genes is not related to intrinsic preference, but that in a cell where methylation was stochastically distributed in those regions, the cells do not grow or die off. The authors after all state that the introduction of the 4 DNMTs leads to reduced viability and growth rates. I highly doubt this could explain all their results (i.e. preference for methylation between nucleosomes, motif preferences), but I would be concerned whether some of the observed effects could be due to selective pressures rather than intrinsic constraints on DNMT activity.

I do have several other concerns related to the Hi-C data. First, global changes in cis/trans ratios are often used as a measure of the “signal-to-noise” in a Hi-C experiment (PMID: 30890172). From our experience, these measures can vary from experiment to experiment, and are often correlated across batches. Therefore, I think it is critical that the Hi-C experiments are performed in the same batches (i.e. same day, processed in parallel). If that is what has been done, that is great, and the authors should make that clear. If not they really need to perform these experiments exactly in parallel to show that this is not the result of batch effects.

In addition, some of the other effects they observe (i.e. condensing of centromeres, changes in telomere association) should be readily visibly by microscopy. I think the manuscript would be strengthened if they could show the same effects using something like telomere or centromere FISH or live cell imaging.

Minor points:

I find it interesting that the levels of methylation are considerably higher (27% vs. ~9%) in the saturation vs. exponential synchronized G1. I am hoping the authors could add some additional information to clarify this difference. The simplest explanation in my mind is the length of time that the cells have spent in G1 (i.e. not dividing), but it isn't clear then how long the saturation cells have spent in G1. Is there an estimate they can make of how long the Saturation cells have been not dividing? Similarly, if they extended the length of time the cells are synchronized in G1 (i.e. to 48 hours) do they see a continued increase in methylation levels?

On a related point, the authors write: “The likelihood ratio tests show high significant support for that the presence of multiple components for both samples ($p < 1.0e-16$ in both cases). The presence of two distinct populations of reads in the samples in exponential phase suggests that the DNA methylation maintenance machinery may be not fully functional, leading to a difference in average methylation between the original and daughter DNA strands.” Can they test this explicitly, i.e. by performing similar analysis of Nanopore data from cells isolated according to cell cycle profile during exponential growth? The prediction would be that by identifying cells in G1 vs. S/G2/M you could resolve the potential mixture of populations. These results are quite interested related to the “intrinsic” ability of the maintenance methyltransferase ability to maintain DNA methylation through DNA replication.

This is a naïve question, but do yeast have CpG islands? If so, can the authors check whether these would be depleted of methylation as is seem in mammals?

For Figure 1a, there is no Y-axis to show the level of methylation.

I don't think the data they have is strong enough to support the following statement: "suggesting that H3K4 and H3K36 methylation are capable of tightly controlling DNA methylation even upon DNMTs overexpression by a direct mechanism." I think to really say that the modifications are controlling the patterns of DNA methylation you need to do these experiments in mutants that lack the modifications.

It is really interesting that the introduction of the 4 DNMTs leads to the generation of "fuzzier" nucleosome patterns. My only criticism here is that the results are only listed in a supplemental table. They should list this in the main text/show some examples as I think this is a particularly interesting point as it suggests that methylation patterns may intrinsically affect nucleosome positioning in vivo.

Some typos:

In the intro, reference 30 is repeated twice at the same instance: "To determine the intrinsic impact of DNA methylation (i.e. that independent of specific methylation recognition machinery) on genome organization, we use budding yeast as a model system. *S. cerevisiae*'s genome is deprived of any cytosine methylation 30 30".

"These results agree with a bulk of in vivo 25, 26, 27, 28, in vitro and in silico data" should be "These results agree with the bulk..."

Reviewer #3 (Remarks to the Author):

The manuscript entitles "Impact of DNA methylation on 3D genome structure" by Buitrago et al investigate the effect of DNA methylation on 3D genome structure. Expression of DNA methyltransferase (DNMT) in a model organism without any endogenous DNA methylation can provide information about chromatin specificity. Expression of DNMT in budding yeast have been reported previously, in both *S. cerevisiae* or *K. phaffii* (new name of *Pichia pastoris*). In *S. cerevisiae*, initial studies in which less than 0.5% of cytosines became methylated showed enzymatic activity of DNMT in budding yeast, and the requirement for N-terminal region of H3 for targeting DNA. Study from Pellegrini labs achieved 3 to 7% methylation in *S. cerevisiae*, leading to genome wide analysis of methylation pattern. Human DNMT were expressed in *K. phaffii*, and provide some additional information on effect of methyltransferase on S-adenosyl methionine (SAM) metabolism.

Here, authors achieved higher expression level of DNMT, associated with a significant fraction of genomic DNA modification. This achievement allow to reach interesting conclusions. With such high level of methylation, authors could generate a large amount of high quality dataset:

DNA was homogeneously methylated in stationary phase, but not in exponentially growing cells.

Testing expression of 3 out of 4 DNMT, authors could decipher that DNMT3a and 3b have sequence

specificity, but not DNMT1 and DNMT3L.

Methylation occurs preferentially at linker DNA, and NFR.

Authors could also observe clear transcriptional impacts, both up and down, showing that independently of any methyl DNA binding proteins, physical properties of methylated DNA impacts transcription.

Furthermore, authors focused on high-order chromatin organization using HiC, which is clearly interesting and important. With appropriate control, this study could allow for the first time to evaluate function of DNA methyltransferase on chromosome conformation independently of any methyl DNA binding proteins.

Major comments

A major breakthrough of this study is the analysis using HiC of chromosome reorganization following DNMT expression. Unfortunately, using asynchronous culture, DNMT expression lead to delay in G2 phase of cell cycle. Recent study using either HiC, or MicroC, showed a considerable reorganization of contact matrix along chromosomes from G1 to G2 phase of yeast cell cycle (see for example costantino, Biorxiv, 2020 - DOI: 10.1101/2020.06.11.146902). In G1, chromosome are organized as small chromosome interacting domain (CID). In G2, cohesin dependent loops are organizing chromosomes, associated with decrease inter-chromosomal versus increase intra-chromosomal contact. Reorganization observed here could come from DNMT expression, or at least in part bias toward G2 in asynchronous yeast cell culture. A specific figure should be dedicated to cell cycle stage of cells used to performed HiC analysis. Authors should evaluate if DNMT re-organisation documented here is clearly different than documented G2 conformation. Ideally, as for methylation level authors could use synchronized cell culture to investigate if DNMT are contributing to cohesin dependent loop size or position.

Authors observed a very specific transcriptional response correlated with URS1, and binding site for Ume6, a subunit of Rpd3 complex. Albeit interesting, this is only a correlation. It remains to be shown that binding of such transcription factor is directly affected by methylation. Working with a genetically amenable model, authors could have establish a causal link by 1/ performing ChIP of Ume6 and 2/comparing transcriptional response of DNMT expression in ume6 deletion, or rpd3 mutant.

Minor comments

HiC map should be shown, and not only log ratio.

Authors propose that stiffness of chromatin is increased by methylation. Direct validation of such proposal could be achieved by measuring either chromatin motion, or compaction by live cell imaging in vivo.

Dear reviewers,

We would like to thank you for your comments that have helped us improve our work on the effect of DNA methylation on chromatin structure and 3D genome organisation. We have tried to address all of your concerns, either by performing new experiments or by reformulating some of the discussion, and we hope that you will find the revised document acceptable.

Please find below the detailed answers to your comments.

Reviewer #1 (Remarks to the Author):

Comments for 262488_0_merged_1592425197

The manuscript entitled "Impact of DNA methylation on 3D genome structure" by Buitrago D., Labrador M., et al. addresses the relationship between the DNA methylation mark per se and chromatin function/architecture. The authors were able to reconstruct the mammalian methylation machinery expressing four murine proteins in yeast: de novo DNA Methyltransferases (DNMT3a and DNMT3b), maintenance DNA Methyltransferase (DNMT1) and a non-catalytic accessory factor (DNMT3L). The authors were able to achieve higher levels of 5meC than previous studies and they were able to recapitulate the results found in literature, confirming the validity of the model. Their main findings are:

- Introduction of DNA methylation in yeast generates a similar pattern as mammalian cells upon expression of two de novo DNA Methyltransferases, a maintenance methyltransferase and an accessory factor. They were able to determine a DNMT3-specific intrinsic sequence specificity.
- They recapitulated previous findings that DNA Methylation is mainly found between nucleosomes and it can influence nucleosome positioning, transcription factor binding and gene expression.
- DNA Methylation induces an increase in intra-chromosome contacts while decreasing inter-chromosome contacts, suggesting that RNA pol III transcribed loci can be involved in the organization.

Comments:

1) My main concern about the manuscript is that the authors do not consider that the presence of the DNA Methyltransferases themselves (2 de novo, 1 accessory factor, and 1 maintenance) could cause nucleosome fuzziness and/or rearrangements of the 3D genome structure, which are two of their main findings. It is well-established in the literature that DNMT3a/b/L interact with histone tails and they might have multimerization potential. The maintenance DNMT1 can also be considered as a reader of hemimethylated CpG sites. A possible solution would be to express in yeast:

- A catalytic dead version for both active de novo DNMT3s (point mutant), in

order to rule out the above mentioned DNMT3-dependent effect;
- the bacterial CpG methyltransferase M.SssI, if they want to understand the effects of 5meC per se. The MTase should not be affected by histone modifications, but it should be blocked by the presence of nucleosomes;

We thank the reviewer for their relevant comment. Following their advice, we performed the suggested experiment and expressed the catalytic dead version of the DNMTs, i.e. Dnmt3b^{P656V/C657D}, Dnmt3a^{P705V/C706D} and DNMT1^{C1229S} described in Takebayashi et al (2007) and Nowialis et al (2019)). We first performed some site directed mutagenesis to insert the relevant mutations. We then co-expressed the three mutated DNMTs together with the cofactor DNMT3L in yeast, performing MNase-seq experiments to compare the nucleosome fuzziness between the control strain containing the empty vectors, the strain expressing the active DNMTs, and the strain expressing the catalytically dead DNMTs. As shown in Supplementary figure S7A, the 2 replicas with the catalytically dead DNMTs segregate with the control samples and not with the samples expressing active DNMTs. Similarly, looking at nucleosome fuzziness, we can observe that the two replicas with the catalytically inactive DNMTs have a lower proportion of fuzzy nucleosomes than the replicas having active DNMTs (Supplementary figure S7B). These results strongly suggest that the effects on nucleosome positioning observed in the methylated samples are mostly due to the DNA methylation. Results are briefly summarized and commented in the revised version of the paper, and a new Supplementary figure (S7) has been added.

2) A second point is that, based on the emphasis given in the title, I expected more data on genome structure. I suggest performing experiments with an orthogonal technique to validate the HiC results on a few loci that show differences between the control and the DNA methylation strain. In an ideal world, it would be through FISH/CRISPR-imaging, SPRITE or GAM.

This is again a relevant comment. The validation of Hi-C results is indeed an important aspect of any genome structure study. As suggested by the reviewer, SPRITE and GAM would be two very good options but as far as we know, these techniques have not been used in yeast, due to the small size of its nuclei. Thus, in order to perform the validation suggested by the reviewer, we took advantage of the system set up by Bystricky et al, (2012) and previously used by Belton et al, (2015) to validate their Hi-C results. This system contains 3 operator arrays (TetO, LacO and LambdaO) inserted in 3 strategic regions of chromosome III, the 2 silent *MAT* loci (HML and HMR) and the *MAT* locus itself. Using this system, we managed to confirm one of the observations we made using Hi-C, that is to say, that the distance between HML and the *MAT* locus was reduced upon DNA methylation. This result has been incorporated in the manuscript as Supplementary figure S18.

3) Previous manuscripts were able to predict the abundance of DNA methylation or DNMT3 occupancy based on the presence of a few chromatin marks genome-wide. Can the presence of a DNMT3-specific target DNA sequence improve the prediction?

As stated by the reviewer, prediction of epigenomic marks such as DNA methylation can be performed based on existing epigenomic data. For example, Ernst, J and Kellis, M (2015) have developed ChromImpute for large-scale systematic epigenome imputation. However, looking in more details into DNA methylation in differentiated mammalian cells using the same analysis that we performed on the yeast, we did not observe any sequence specificity so we do not think that the sequence context of the CpGs could be a major factor in the prediction of DNA methylation in mammals.

4) Minor comments:

- Add page numbers

Page numbers have been added.

- Abstract:

- DNA methyltransferases, not DNA Methyl Transferases;

This has been corrected.

- *S. cerevisiae* should be in italics

This has been corrected.

- Introduction:

- "...that DNA methylation makes DNA less flexible and..." should be "...that DNA methylation makes the DNA less flexible and...";

This has been corrected.

- in the sentence "S. cerevisiae does not have methylation/demethylation machinery and no methylated DNA binding domain has been characterized", please specify that yeast lacks the DNA methylation machinery, since it possesses other methyltransferases;

This has been modified as suggested by the reviewer.

- The Addition of DNA methylation machinery..., instead of "Addition of methylation machinery...";

This has been corrected.

- Results:

- Supplementary Fig. S2A: how many replicates for each time point/conditions? Please add the information

The growth curve presented in Sup Fig S2A was obtained with one culture from one transformant but is representative of the growth we observed with all the transformants that we tested. The information was added in the legend of the figure.

- It seems that I cannot find the DNA Methylation levels measured by HPLC/MS

A table (Supplementary table S1) has been added in the supplementary material with the DNA methylation level measured by HPLC/MS and the text was modified as follows :

“DNA methylation was first assessed by HPLC/MS, which showed that using this approach, we could reach up to 4.2% of cytosines methylated after 38 hours of induction in cells collected in stationary phase (Supplementary Table S1), and then analysed at single base pair resolution in several independent transformants, using Illumina whole genome bisulfite sequencing (WGBS).”

- Supplementary Figures S3C/S4: I suggest to differentiate the Illumina-WGBS and Nanopore-seq tracks with different colors

The nanopore tracks are now in magenta. The legend of the figures has been changed as follows:

Supplementary Figure S3. Heatmap showing the pairwise CpG methylation correlation in (A) two nanopore replicates and (B) in one nanopore vs one WGBS replicate. (C) Methylation pattern for a 50kb region of chromosome II (50,000-100,000) in WGBS (top 2 tracks **in blue**) and nanopore (bottom tracks **in magenta**) samples for 2 replicates of each condition.

“Supplementary Figure S4. Methylation pattern from WGBS (top tracks **in blue**) and nanopore (bottom tracks **in magenta**) samples for 2 biological replicates at (A) the HML locus, (B) the HMR locus, (C) the rDNA locus and (D) a telomeric region in chromosome IV.”

- For the sentences: “For the cells synchronized in G1 we can see that ~50% of CpG sites have <5% methylation, but there are almost 20% of sites with methylation over 20%, and ~5% of sites with methylation over 50%. For the cells at saturation the methylation levels are generally higher, with 50% of sites having >15% methylation and 25% of sites having methylation over 50%.”, I believe it will be easier for the reader to show an additional plot with the number of sites (or percentages) and the categories of DNA methylation levels you mentioned (<5%; 5-20%; 20-50%; >50%).

The plot suggested by the reviewer 1 has been added as supplementary fig S5C.

- Sentence: “The presence of two distinct populations of reads in the samples in exponential phase suggests that the DNA methylation maintenance machinery maybe not fully functional, leading to a difference in average methylation between the original and daughter DNA strands”. Another hypothesis is that the DNA Methylation machinery is indeed fully functional, but it’s not 100% effective due to the rapid replication of the genomic DNA in yeast during the exponential phase. Please articulate.

This is a very appropriate comment and we modified the text as follow to include this hypothesis.

“The presence of two distinct populations of reads in the samples in exponential phase suggests that the DNA methylation maintenance machinery may be not fully functional or that the replication phase in yeast is too short for the mammalian maintenance machinery, leading to a difference in average methylation between the original and daughter DNA strands.”

- Figure 1A,B: Please indicate that the plots are showing DNA Methylation levels from “replicating cells, synchronized in G1”. Please add the y-axis (methylation levels/fraction).

Figures 1A, B have been corrected and the y-axis was added.

- Figure 1G: Please differentiate with colors the stationary vs. G1 tracks.

Figure 1G has been modified and the G1 tracks are now in green (the stationary tracks are in blue).

- For the DNA methylation and Nucleosome positioning section, it would be great to cite NOME-seq, an experimental procedure to map DNA Methylation and nucleosomes on the same molecule.

We corrected this omission and cited NOME-seq in the section of the discussion about DNA methylation and Nucleosome positioning.

- Figure 5A: maybe specify in the caption that the methylation probability is the shaded area. I would also set 0.4 as the maximum for the 5mC y-axis to zoom into the signal a little bit

Figure 5A and its caption have been modified as suggested by the reviewer.

- Figure 5B/C and section discussing DNA Methylation and transcription. The authors state “A differential expression analysis (Fig. 5B) shows that genes which are very lowly methylated do not change their expression level between the control and transformed cells, while high methylation levels lead to important changes in gene activity in both directions: towards greater and lower expression (Fig. 5C and Supplementary table S1).” The changes in gene expression could also be due to the presence of the DNA methyltransferases themselves on the chromatin, not only to the DNA methylation mark alone. Please take this into account and modify the sentence accordingly. You can just state that there is a trend/correlation.

As stated by the reviewer, we cannot rule out an effect of the DNA methyltransferase recruitment on the chromatin. The text has been modified as follow to include this possibility in our interpretation of the results.

“Although we cannot discard any indirect effect due to the binding of the DNMTs on the chromatin, we see a very strong correlation between gene expression and methylation level for a subset of genes involved in meiosis and that appear to share a

common sequence in their regulatory region (Fig. 5C, D), corresponding to the binding site of Ume6p, a subunit of the histone deacetylase complex Rpd3p known to repress early meiotic gene expression."

- Figure S9: You should also plot, as a control, the methylation and expression fold changes for a set of genes regulated by a specific transcription factor (non-overlapping with Ume6p) and one for randomly picked genes (a subset or all).

Our data did not permit us to produce the plots suggested by the reviewer. As shown in fig 5C, we only observed 20 downregulated genes and 63 upregulated genes and we could not find another subset of genes (non-overlapping with Ume6p), that were regulated by a specific transcription factor that would show any significant changes of expression. For example, none of the targets of Gcn4p, Cin5p or Hac1p, which also contain a CpG in their binding site, showed changes of expression.

- Discussion:

- The authors state "Again, these differences are intrinsic and not coupled to any specific directing mechanism, which demonstrate the intrinsic ability of methylation to alter cellular phenotype.": as mentioned by the authors and other reports, the underlying chromatin structure (presence of nucleosomes and histone modifications) dictates the possible substrates for the DNA Methyltransferases. It is correct that there are no factors that are specifically directing DNMT3s to a specific sequence. Please clarify the meaning of "which demonstrate the intrinsic ability of methylation to alter cellular phenotype".

We have revised this section of the discussion and the phrase that the referee was referring to has been modified as follows :

"Again, these differences are intrinsic and not coupled to any specific directing mechanism, which suggests a certain level of specificity of the DNMTs."

Reviewer #2 (Remarks to the Author):

Buitrago et al. report an interesting study on the effect of expression of the mouse DNA methylation machinery in yeast. The goal of this approach is to study the "intrinsic" preferences of the DNA methylation machinery (and their effects) on genome structure and function. In this sense, the overall approach seems most reminiscent of past work exploring in vitro nucleosome position and DNA methylation patterns, but this has the added advantage that this study is carried out in vivo and this can be used to study the relationship with other biological processes in the nucleus. The authors find that the introduction of the 4 DNMT proteins from mice leads to increased levels of DNA methylation (dependent upon conditions), changes in nucleosome patterning, 3D genome structure and some gene expression alterations. Further they characterize the patterns of DNA methylation deposition and correlate these with features the genome, such as the status of genes. Overall, I think that study will invite a criticism of lacking physiological relevance, but I think this misses the point, as I

think the most clear comparison for this study have been past in vitro work. I find the manuscript interesting and well written. However, there are still several concerns that I have that I hope the authors can address before I think it would be appropriate for publication in Nature Communications.

We thank the reviewer for their positive general comment on our manuscript. Detailed answers to specific comments can be found below.

Major points:

Perhaps the single biggest concern that I have is that while I find the data compelling, I am concerned that the patterns that they observe could be the result of growth/selection instead of innate preferences of the DNA methyltransferase activity. In other words, perhaps the observation of a low level of DNA methylation near the promoters of genes is not related to intrinsic preference, but that in a cell where methylation was stochastically distributed in those regions, the cells do not grow or die off. The authors after all state that the introduction of the 4 DNMTs leads to reduced viability and growth rates. I highly doubt this could explain all their results (i.e. preference for methylation between nucleosomes, motif preferences), but I would be concerned whether some of the observed effects could be due to selective pressures rather than intrinsic constraints on DNMT activity.

We understand the reviewer's concern and even if we cannot totally rule out the possibility that some kind of growth selection is being applied, the fact that the methylation pattern does not change over time suggests that this pattern is not the result of selection. Even as early as 6 hours after induction, we already observe lower methylation at promoters increasing toward the 3' end of the genes. This concern is commented in the revised version of the manuscript as follows :

"While we cannot rule out this pattern being due to selection (i.e., that cells with no methylation at promoters have a selective advantage), the fact that the pattern is established very early on (6 hours after the induction of the DNMTs) and remains stable across all time points indicates that this has a minor effect, if any."

I do have several other concerns related to the Hi-C data. First, global changes in cis/trans ratios are often used as a measure of the "signal-to-noise" in a Hi-C experiment (PMID: 30890172). From our experience, these measures can vary from experiment to experiment, and are often correlated across batches. Therefore, I think it is critical that the Hi-C experiments are performed in the same batches (i.e. same day, processed in parallel). If that is what has been done, that is great, and the authors should make that clear. If not they really need to perform these experiments exactly in parallel to show that this is not the result of batch effects.

We thank the reviewer for raising this important point. All the Hi-C experiments (2 control and 2 methylated samples) were done on the same batches. In addition, we used the same cell cultures to do the Hi-C, the WGBS, the RNAse-seq and the MNase-seq to make sure the data could be correlated. Only the nanopore sequencing was done on a separate culture because it was added later to the project. We added this information in the following sentence :

“We performed Hi-C experiments in control and methylated populations at saturation *in two replicas processed in parallel* to explore the intrinsic effect of DNA methylation in the global chromatin structure.”

In addition, some of the other effects they observe (i.e. condensing of centromeres, changes in telomere association) should be readily visibly by microscopy. I think the manuscript would be strengthened if they could show the same effects using something like telomere or centromere FISH or live cell imaging.

We tried to look at telomere association by immunofluorescence using an antibody against RAP1, but data were rather noisy. However, we managed to validate some of our Hi-C results using the system set up by Kirstin Bystricky (Bystricky et al, 2012) and previously used in Belton et al, (2015) to validate their Hi-C results. This system contains 3 operator arrays (TetO, LacO and LambdaO) inserted in 3 strategic regions of chromosome III, the 2 silent *MAT* loci (HML and HMR) and the *MAT* locus itself. Using this system, we managed to confirm the Hi-C result, that is to say, that the distance between HML and the *MAT* locus was reduced upon DNA methylation. This result has been incorporated and discussed in the manuscript as Supplementary figure S18.

Minor points:

I find it interesting that the levels of methylation are considerably higher (27% vs. ~9%) in the saturation vs. exponential synchronized G1. I am hoping the authors could add some additional information to clarify this difference. The simplest explanation in my mind is the length of time that the cells have spent in G1 (i.e. not dividing), but it isn't clear then how long the saturation cells have spent in G1. Is there an estimate they can make of how long the Saturation cells have been not dividing? Similarly, if they extended the length of time the cells are synchronized in G1 (i.e. to 48 hours) do they see a continued increase in methylation levels?

This is a very interesting comment. We performed flow cytometry to monitor the cells during 48 hours from the time we induced the expression of the DNMTs to the time we collected the cells. The results are presented in Supplementary fig S2D and reveal that cells spent between 24 and 36 hours in G1 without dividing. This information was added in the following sentence.

“This was done both for cells in exponential growth phase synchronized in G1, and for cells at saturation that spent between 24 and 36 hours in G1 without dividing (Supplementary fig S2D).”

On a related point, the authors write: “The likelihood ratio tests show high significant support for that the presence of multiple components for both samples ($p < 1.0e-16$ in both cases). The presence of two distinct populations of reads in the samples in exponential phase suggests that the DNA methylation maintenance machinery may be not fully functional, leading to a difference in average methylation between the original and daughter DNA strands.” Can they test this explicitly, i.e. by performing similar analysis of Nanopore data from cells isolated according to cell cycle profile during exponential growth? The prediction would be that by identifying cells in G1 vs. S/G2/M you could resolve the potential mixture of populations. These results are quite interested related to the “intrinsic” ability of the maintenance methyltransferase ability to maintain DNA methylation through DNA replication.

We looked into performing the experiment suggested by the reviewer, however preliminary WGBS data we have from cultures that were followed for 1 cycle after being synchronized in either G1/S or G2/M were inconclusive. Taking this into account, as well as the comments from the first reviewer, we have modified the section in the text to make it clear that we cannot test between the different hypotheses that could explain these results.

This is a naïve question, but do yeast have CpG islands? If so, can the authors check whether these would be depleted of methylation as is seen in mammals?

Saccharomyces cerevisiae’s genome does not have CpG islands. The frequency of CpG in the yeast genome is slightly depleted when compared with GpC (2.9% vs 3.7%), but this depletion is much less than in humans (1.0% vs 4.2%) with a fairly even distribution of CpGs over the genome and the average frequency of CpGs in yeast being close to that seen within vertebrate CpG islands.

For Figure 1a, there is no Y-axis to show the level of methylation.

Figures 1A has been corrected and the y-axis was added.

I don’t think the data they have is strong enough to support the following statement: “suggesting that H3K4 and H3K36 methylation are capable of tightly controlling DNA methylation even upon DNMTs overexpression by a direct mechanism.” I think to really say that the modifications are controlling the patterns of DNA methylation you need to do these experiments in mutants that lack the modifications.

We agree with the reviewer that our experiments cannot confirm the role of H3K4 and H3K36 in DNA methylation. However, previous studies using yeast as a model organism have shown that DNA methylation was extremely reduced in SET mutants (see Morselli et al. (2015) and Hu et al. (2009)) showing that these histone marks played an important role in DNA methylation in their system. Our ChIP-seq experiment was therefore a confirmation of the previously reported correlation between H3K4 and H3K36 methylation and DNA methylation.

It is really interesting that the introduction of the 4 DNMTs leads to the generation of “fuzzier” nucleosome patterns. My only criticism here is that the results are only listed in a Supplementary table. They should list this in the main text/show some examples as I think this is a particularly interesting point as it suggests that methylation patterns may intrinsically affect nucleosome positioning in vivo.

We thank the reviewer for their comment and we have moved the results table into the main text as suggested. This table is now the Table 2.

Some typos:

In the intro, reference 30 is repeated twice at the same instance: “To determine the intrinsic impact of DNA methylation (i.e. that independent of specific methylation recognition machinery) on genome organization, we use budding yeast as a model system. *S. cerevisiae*'s genome is deprived of any cytosine methylation 30 30”.

The text was corrected accordingly.

“These results agree with a bulk of in vivo 25, 26, 27, 28, in vitro and in silico data” should be “These results agree with the bulk...”

The text was corrected accordingly.

Reviewer #3 (Remarks to the Author):

The manuscript entitles “Impact of DNA methylation on 3D genome structure” by Buitrago et al investigate the effect of DNA methylation on 3D genome structure. Expression of DNA methyltransferase (DNMT) in a model organism without any endogenous DNA methylation can provide informations about chromatin specificity. Expression of DNMT in budding yeast have been reported previously, in both *S. cerevisiae* or *K. phaffii* (new name of *Pichia pastoris*). In *S.*

cerevisiae, initial studies in which less than 0.5% of cytosines became methylated showed enzymatic activity of DNMT in budding yeast, and the requirement for N-terminal region of H3 for targeting DNA. Study from Pellegrini labs achieved 3 to 7% methylation in *S. cerevisiae*, leading to genome wide analysis of methylation pattern. Human DNMT were expressed in *K. phaffii*, and provide some additional information on effect of methyltransferase on S-adenosyl methionine (SAM) metabolism.

Here, authors achieved higher expression level of DNMT, associated with a significant fraction of genomic DNA modification. This achievement allow to reach interesting conclusions. With such high level of methylation, authors could generate a large amount of high quality dataset:

DNA was homogeneously methylated in stationary phase, but not in exponentially growing cells.

Testing expression of 3 out of 4 DNMT, authors could decipher that DNMT3a and 3b have sequence specificity, but not DNMT1 and DNMT3L.

Methylation occurs preferentially at linker DNA, and NFR.

Authors could also observe clear transcriptional impacts, both up and down, showing that independently of any methyl DNA binding proteins, physical properties of methylated DNA impacts transcription.

Furthermore, authors focused on high-order chromatin organization using HiC, which is clearly interesting and important. With appropriate control, this study could allow for the first time to evaluate function of DNA methyltransferase on chromosome conformation independently of any methyl DNA binding proteins.

Major comments

A major breakthrough of this study is the analysis using HiC of chromosome reorganization following DNMT expression. Unfortunately, using asynchronous culture, DNMT expression lead to delay in G2 phase of cell cycle. Recent study using either HiC, or MicroC, showed a considerable reorganization of contact matrix along chromosomes from G1 to G2 phase of yeast cell cycle (see for example costantino, Biorxiv, 2020 - OI: 10.1101/2020.06.11.146902). In G1, chromosome are organized as small chromosome interacting domain (CID). In G2, cohesin dependent loops are organizing chromosomes, associated with decrease inter-chromosomal versus increase intra-chromosomal contact. Reorganization observed here could come from DNMT expression, or at least in part bias toward G2 in asynchronous yeast cell culture. A specific figure should be dedicated to cell cycle stage of cells used to performed HiC analysis. Authors should evaluate if DNMT re-organisation documented here is clearly different than documented G2 conformation. Ideally, as for methylation level authors

could use synchronized cell culture to investigate if DNMT are contributing to cohesin dependent loop size or position.

As stated by the reviewer, several very nice studies have shown that 3D genome organisation changes drastically depending on the cell cycle stages and this effect has to be taken into account in any Hi-C study. In our study, all the experiments were performed either in cells synchronized in G1 in the case of exponentially growing cells, or in cells arrested in G0. This was confirmed by flow cytometry where we see that cells were all in G1 like phase when they were collected . We added this result in Supplementary Figure S2D and we have made this point more clear in the revised text.

Authors observed a very specific transcriptional response correlated with URS1, and binding site for Ume6, a subunit of Rpd3 complex. Albeit interesting, this is only a correlation. It remains to be shown that binding of such transcription factor is directly affected by methylation. Working with a genetically amenable model, authors could have establish a causal link by 1/ performing CHIP of Ume6 and 2/comparing transcriptional response of DNMT expression in *ume6* deletion, or *rpd3* mutant.

The reviewer is again correct: we did not prove a direct effect of DNA methylation on Ume6p binding to DNA. To test this, and as suggested by the reviewer, we performed CHIP-seq of Ume6 in control and methylated cells (data not shown). However, we could not see a clear difference between the two set of samples. This result could be explained by the fact that the UME6 sites were not homogeneously methylated in all cells (as shown in Supplementary table S4 and Supplementary Figure S10a) so if Ume6p binding is not totally inhibited, the difference might not be picked up by CHIP-seq experiment. Therefore, we opted to use an *in vitro* approach, producing recombinant Ume6 in *E. coli* and performing a gel shift assay using an unmethylated or methylated probe. This experiment showed that Ume6p has a lower affinity for the methylated probe than for the unmethylated one. The result has been added in the Supplementary fig S10B_C and is referred in the text as follows :

“This was further confirmed *in vitro* by showing that recombinant Ume6p has higher affinity for an unmethylated URS1 site than for a methylated one (Supplementary fig S10B-C)”

Minor comments

HiC map should be shown, and not only log ratio.

The 4 Hi-C maps were added in Supplementary fig S12

Authors propose that stiffness of chromatin is increased by methylation. Direct validation of such proposal could be achieved by measuring either chromatin motion, or compaction by live cell imaging *in vivo*.

While we agree with the reviewer that it would be interesting to measure chromatin motion or compaction by live cell imaging *in vivo* and this is something we should definitely try to set up in the future, it was not possible to do this within the timeframe of the project. However, we did perform live cell imaging and compared cells with and without induction of the DNMTs using a system set up by Bystricky et al, (2012) and previously used by Belton et al, (2015) to validate their Hi-C results. It consists of a strain containing 3 operator arrays (TetO, LacO and LambdaO) inserted in 3 strategic regions of chromosome III, the 2 silent *MAT* loci (HML and HMR) and the *MAT* locus itself. Using this system, we observed that the distance between the HML and Mat loci were reduced in the methylated cells, confirming our observations about chrIII structural changes (Supplementary figure S18). It would be interesting to monitor those changes over time while DNA methylation is set up, but the length of the process (48 to 72 hours) to reach a high level of methylation is such that it would require a substantial amount of time to set up.

The numeration of the pre-existing tables and figures was modified in the text to take into account the new figures. Also, the material and methods describing the experiments that were performed to answer the reviewer's comments were included in the revised version of the manuscript.

Yours,

Isabelle Brun Heath (co-corresponding author).

REVIEWERS' COMMENTS

Reviewer #1 (Remarks to the Author):

The authors have adequately addressed all of my previous comments

Reviewer #2 (Remarks to the Author):

The authors have sufficiently address the concerns that I have raised in the prior round of review.

Reviewer #3 (Remarks to the Author):

The revised manuscript entitles “Impact of DNA methylation on 3D genome structure ” by Buitrago et al investigate the effect of DNA methylation on 3D genome structure. Expression of DNA methyltransferase (DNMT) in a model organism without any endogenous DNA methylation can provide informations about chromatin specificity.

Here, authors achieved higher expression level of DNMT, associated with a significant fraction of genomic DNA modification. Authors could also observe clear transcriptional impacts, both up and down, showing that independently of any methyl DNA binding proteins, physical properties of methylated DNA impacts transcription.

Furthermore, authors focused on high-order chromatin organization using HiC, which is clearly interesting and important. In this revised version, this study could allow for the first time to evaluate function of DNA methyltransferase on chromosome conformation independently of any methyl DNA binding proteins. Authors have successfully addressed my comments,